# A New Approach to Controlling Linear Dynamical Systems

**Anand Brahmbhatt**[1][*]   **Gon Buzaglo**[1][*]   **Sofiia Druchyna**[1][*]   **Elad Hazan**[1,2][*]

[1]Computer Science Department, Princeton University
[2]Google DeepMind Princeton

## Abstract

We propose a new method for controlling linear dynamical systems under adversarial disturbances and cost functions. Our algorithm achieves a running time that scales polylogarithmically with the inverse of the stability margin, improving upon prior methods with polynomial dependence maintaining the same regret guarantees. The technique, which may be of independent interest, is based on a novel convex relaxation that approximates linear control policies using spectral filters constructed from the eigenvectors of a specific Hankel matrix.

## 1 Introduction

Controlling linear dynamical systems (LDS) under adversarial disturbances is a core challenge at the interface of control and online learning, with direct relevance to reinforcement learning, robotics, and sequential decision making. Formally, given control inputs $\mathbf{u}_t \in \mathbb{R}^n$ and disturbances $\mathbf{w}_t \in \mathbb{R}^d$, the state $\mathbf{x}_t \in \mathbb{R}^d$ evolves as

$$\mathbf{x}_{t+1} = A\mathbf{x}_t + B\mathbf{u}_t + \mathbf{w}_t \,. \tag{1.1}$$

At each step a convex cost $c_t(\mathbf{x}_t, \mathbf{u}_t)$ is revealed, and the learner must choose controls $\mathbf{u}_t$ to minimize cumulative loss.

The classical Linear Quadratic Regulator (LQR) assumes quadratic costs and i.i.d. stochastic noise Kalman (1960). These assumptions rarely hold in modern applications, where disturbances and costs may be structured, correlated, or even adversarial. Consider flying a drone in the wild: the dynamics $(A, B)$ follow physics, but disturbances such as wind or malfunction are unpredictable. The cost captures both the control objective and the environment, and our immediate actions must adapt over time. For instance, reaching a target location requires adjusting turns on the fly in response to other drones, birds, trees, or buildings. Such scenarios call for a more general and robust model of control. We therefore study the fully adversarial setting, with known time-invariant dynamics, full state observation, and convex Lipschitz costs, commonly referred to as *online control* Hazan & Singh (2025).

**Minimizing Regret in Control.**   In adversarial settings, minimizing cumulative cost is generally intractable, as the decision maker only observes the loss after choosing the control input. Instead, following the standard approach in online decision making, we aim to compete with the best stationary policy $\pi \in \Pi$ in hindsight. This is captured by the notion of *regret*, defined as

$$\text{Regret}_T(\mathcal{A}, \Pi) = \sum_{t=1}^{T} c_t\left(\mathbf{x}_t^{\mathcal{A}}, \mathbf{u}_t^{\mathcal{A}}\right) - \min_{\pi \in \Pi} \sum_{t=1}^{T} c_t\left(\mathbf{x}_t^{\pi}, \mathbf{u}_t^{\pi}\right),$$

where $\mathbf{x}_t^{\mathcal{A}}$ is the state induced by the algorithm's controls $\mathbf{u}_t^{\mathcal{A}}$, and $\mathbf{x}_t^{\pi}$ is the state under the fixed policy $\pi$.

A natural choice for $\Pi$ is the class of linear state-feedback policies $\mathbf{u}_t = K\mathbf{x}_t$, standard in control. While optimal in certain settings, finding the best such controller is a nonconvex problem. We therefore adopt improper learning, competing with the best linear policy using a broader class.

---

[*]Authors ordered alphabetically. {ab7728,gon.buzaglo,sd0937,ehazan}@princeton.edu

To enable efficient learning and analysis, it is common to impose structure on the comparator class—for instance, strong stability (Cohen et al., 2018) or diagonal strong stability (Agarwal et al., 2019b). Following this line, we define in Definition 3.4 the class of diagonalizably stable policies, which preserves sufficient expressiveness for general LDS control.

**Marginally Stable Comparators:** For linear policies $\mathbf{u}_t = K\mathbf{x}_t$, stability is determined by the spectral radius $\rho$ of the closed-loop matrix $A + BK$, with state evolution $\mathbf{x}_t = \sum_{i=1}^{t}(A + BK)^{i-1}\mathbf{w}_{t-i}$. While small $\rho$ ensures rapid decay of disturbances, many applications favor *marginal stability*, where $\rho = 1 - \gamma$ for small $\gamma$. This regime preserves long-term memory and yields smoother, more energy-efficient control, useful in settings like robotics, thermal systems, and satellite dynamics. See Section E for an example.

## 1.1 OUR CONTRIBUTIONS

**Spectral Representation for Control:** We introduce *Online Spectral Control* (OSC), which compresses past disturbances into a low-dimensional spectral representation. This viewpoint provides a universal feature map for online control: it reduces the problem to regression on compact spectral features, enabling a provably more efficient algorithm that achieves logarithmic dependence on the stability margin while retaining optimal regret guarantees.

**Exponential Runtime Improvement:** We prove that

$$\text{Regret}_T(\text{OSC}) \leq \tilde{O}(\gamma^{-4}\sqrt{T}),$$

where $\gamma$ is the stability margin. The runtime scales only polylogarithmically in $1/\gamma$, improving on the polynomial dependence of GPC (Agarwal et al., 2019a) via fast online convolution (Agarwal et al., 2024a).

| Method | Regret | Time | Disturbances | Costs |
|---|---|---|---|---|
| LQR | $O(1)$ | $O(1)$ | i.i.d | Fixed Known Quadratic |
| Online LQ (Cohen et al., 2018) | $O(\gamma^{-2.5}\sqrt{T})$ | $O(1)$ | i.i.d | Online Quadratic |
| GPC (Agarwal et al., 2019a) | $\tilde{O}(\gamma^{-5.5}\sqrt{T})$ | $O(\gamma^{-1}\log T)$ | Adversarial | Online Convex Lipschitz |
| **OSC (our Algorithm 1)** | $\tilde{O}(\gamma^{-4}\sqrt{T})$ | $O\left(\log^4\left(\gamma^{-1}T\right)\right)$ | Adversarial | Online Convex Lipschitz |

Table 1: Comparison of different control methods. The highlighted row corresponds to our proposed approach. In the regret bounds, we hide polylogarithmic factors by the notation $\tilde{O}(\cdot)$. Our method is the only one to perform in the most general setting with the best running time.

**Empirical Evaluation:** Appendix 5 presents ablations showing that OSC matches or outperforms GPC across a variety of settings, while using far fewer parameters and integrating naturally with nonlinear models.

## 1.2 RELATED WORK

**Control of Dynamical Systems:** Control theory, grounded in deep mathematical foundations and with a long history of practical applications, dates back to self-regulating feedback mechanisms in ancient Greece. The first formal mathematical treatment is attributed to James Clerk Maxwell (Maxwell, 1868). For a historical perspective, see Fernández Cara & Zuazua Iriondo (2003). The problem of stabilizing general dynamical systems has been shown to be NP-hard (Ahmadi, 2016); see Blondel & Tsitsiklis (2000) for a comprehensive survey on the computational complexity of control.

**Linear Dynamical Systems:** Even simple questions about general dynamical systems are intractable. In control, Lyapunov pioneered the use of linearization to analyze local stability of nonlinear systems (Lyapunov, 1992). The seminal work of Kalman (1960) introduced state-space methods and showed that any LDS can be controlled under stochastic assumptions and known quadratic costs.

**Online Stochastic Control:** Early work in the ML community on control focused on the online LQR setting (Abbasi-Yadkori & Szepesvári, 2011; Dean et al., 2018; Mania et al., 2019; Cohen et al., 2019), achieving $\sqrt{T}$ regret with polynomial runtime. A parallel line of work (Cohen et al., 2018) studied online LQR with adversarially chosen quadratic losses, also achieving $\sqrt{T}$ regret. In all of these works, regret is measured against the best linear controller in hindsight.

**Online Nonstochastic Control:** Recent methods for fully adversarial control are surveyed in Hazan & Singh (2025). These approaches typically learn a parameterized mapping from past disturbances to control inputs, with the number of parameters—and hence the runtime—scaling polynomially with the inverse of the stability margin. Our work builds on the setting of Agarwal et al. (2019a), proposing a more efficient algorithm with similar regret guarantees.

Subsequent work has refined the regret bounds under additional assumptions. Agarwal et al. (2019b) established logarithmic regret for strongly convex costs in the presence of stochastic or semi-adversarial noise, while Foster & Simchowitz (2020) derived similar guarantees for known quadratic costs under fully adversarial noise.

Beyond the full-information setting, Simchowitz et al. (2020) extended the framework to partial observation, Minasyan et al. (2022) analyzed adaptive regret in unknown and time-varying systems, and Sun et al. (2023); Suggala et al. (2024) studied online control with bandit feedback.

Online control has also seen applications in diverse domains, including mechanical ventilation (Suo et al., 2022), meta-optimization (Chen & Hazan, 2023), and the regulation of population dynamics (Golowich et al., 2024; Lu et al., 2025).

**Online Convex Optimization:** Our method reduces online control to online convex optimization. For background on regret minimization and online learning, see Cesa-Bianchi & Lugosi (2006); Hazan (2016).

**Learning in Linear Dynamical Systems:** Hardt et al. (2016) showed that unknown linear dynamical systems (LDSs) can be learned by applying random Gaussian inputs and optimizing via gradient descent. This approach was later extended to higher-dimensional and marginally stable systems by Sarkar & Rakhlin (2019). More recently, Bakshi et al. (2023a) introduced tensor-based methods for learning LDSs, and Bakshi et al. (2023b) extended these techniques to handle mixtures of LDSs, following the formulation of Chen & Poor (2022). While these tensor-based approaches avoid dependence on the system's condition number, their computational complexity still scales with the hidden dimension—a limitation addressed by recent developments in spectral filtering.

**Spectral Filtering:** Spectral filtering was originally introduced for sequence prediction in online learning, as a means of bypassing the non-convexity inherent in learning linear dynamical systems (Hazan et al., 2017; Agarwal et al., 2024b). Hazan et al. (2018) extended the method to systems with non-symmetric dynamics, though their analysis incurred a dependence on the hidden dimension. This limitation was subsequently addressed by Marsden & Hazan (2025), who incorporated autoregressive structure and used Chebyshev polynomial approximations to obtain dimension-free bounds.

While spectral filtering has proven effective in prediction settings, its application to control remains limited. To our knowledge, the only prior use is by Arora et al. (2018), who applied spectral filtering as a black-box subroutine in the offline LQR setting with unknown dynamics. In contrast, we address the online control problem, and develop a new spectral filtering approach applicable to *any controllable* system (Definition 3.1), competing with the best diagonalizably stable policy in hindsight (Definition 3.4).

The key technical difference is that, unlike in prediction settings where past responses are available, online control requires leveraging system stability and integrating over a different set of eigenvalues, resulting in a distinct Hankel structure (see (2.1)).

## 2 ALGORITHM AND MAIN RESULT

Our approach employs a convex relaxation of the linear control problem by computing universal spectral filters and then learning a linear combination of the multiplication of past disturbances with these filters, as described in detail in Algorithm 1. The spectral filters are universal in the sense that they are independent of the specific linear dynamical system (LDS) at hand, the initial state, the disturbances, and the cost functions. In fact, they correspond exactly to the top eigenvectors of the matrix $H \in \mathbb{R}^{m \times m}$ for some memory $m$, whose entries are given by

$$H_{ij} = \frac{(1-\gamma)^{i+j-1}}{i+j-1}, \tag{2.1}$$

---

**Algorithm 1** Online Spectral Control Algorithm

---

1: **Input:** Horizon $T$, number of parameters $h$, memory $m$, step size $\eta$, convex constraints set $\mathcal{K} \subseteq \mathbb{R}^{h \times n \times d}$.
2: Compute $\{(\sigma_j, \phi_j)\}_{j=1}^{h}$, the top $h$ eigenpairs of $H$ from Eq. (2.1).
3: Initialize $M_{1:h}^0 \in \mathbb{R}^{h \times n \times d}$.
4: **for** $t = 0, \ldots, T - 1$ **do**
5:     Define $\tilde{W}_{t-1:t-m} = [\mathbf{w}_{t-1}, \ldots, \mathbf{w}_{t-m}] \in \mathbb{R}^{d \times m}$
6:     Compute control $\mathbf{u}_t = \sum_{i=1}^{h} \sigma_i^{1/4} M_i^t \tilde{W}_{t-1:t-m} \phi_i$
7:     Observe the new state $\mathbf{x}_{t+1}$ and record $\mathbf{w}_t = \mathbf{x}_{t+1} - A\mathbf{x}_t - B\mathbf{u}_t$.
8:     Set $M_{1:h}^{t+1} = \Pi_{\mathcal{K}} [M_{1:h}^t - \eta \nabla_M \ell_t (M_{1:h}^t)]$
9: **end for**
10: **return** $M_{1:h}^T$

---

where $\gamma$ is an assumed bound on the system's instability margin, formally defined in Definition 3.4. We illustrate the filters in Figure 1. The number of required eigenvectors, denoted by $h$, also determines the number of learnable parameters.

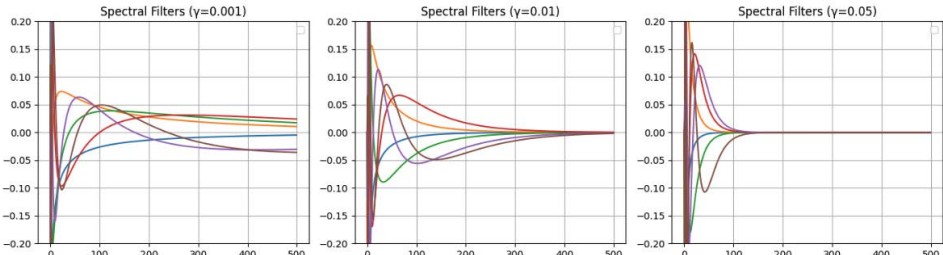

Figure 1: Entries of the first six eigenvectors of $H_{500}$, plotted coordinate-wise.

## 2.1 ALGORITHM

In Algorithm 1 we learn a mapping from disturbances $\mathbf{w}_t$ to controllers $\mathbf{u}_t$. However, instead of learning a linear mapping of the disturbances, we learn a linear mapping of a specific matrix product involving the disturbances and the filters. Notably, in line 7 we explicitly leverage our knowledge of the system to compute the disturbances from the observed state and the chosen controller.

Finally, Algorithm 1 is simply an instance of Projected Online Gradient Descent (Zinkevich, 2003), applied to some convex set $\mathcal{K}$ in which each element is a sequence of $h$ matrices from $\mathbb{R}^{n \times d}$, and the loss functions are the specific memory-less loss functions $\ell_t$, as in Definition 4.4.

## 2.2 MAIN RESULT

While we defer the formalization of our assumptions and notations to Section 3, we present our main result here:

**Theorem 2.1** (Main Theorem). *Let $c_t$ be any sequence of convex Lipschitz cost functions satisfying Assumption 3.3, and let the LDS be controllable (Definition 3.1) and satisfy Assumption 3.2. Then, Algorithm 1 achieves the following regret bound:*

$$Regret_T (\text{OSC}, \mathcal{S}) = \frac{C_0 C_1 \sqrt{T}}{\gamma^4} \log^3 \left( \frac{C_1 T d}{\gamma^3} \right),$$

*where $\mathcal{S}$ is the class of linear policies defined in Definition 3.4. This result holds under the following choice of inputs:*

*(i)* $m = \left\lceil \frac{1}{\gamma} \log \left( \frac{8 C_1 \sqrt{T}}{\gamma^3} \right) \right\rceil$,

*(ii)* $h = \left\lceil 4 \log T \log \left( \frac{900 C_1 dT}{\gamma^3} \right) \right\rceil$,

*(iii)* $\eta = C_2 \sqrt{\frac{\gamma^3}{Tmh}}$,

*(iv)* $K$ is the set from Definition 4.2,

*where the constants are defined as follows:*

$$C_0 \leq 10^3, \quad C_1 = G \kappa_B \kappa^8 W^2, \quad C_2 = \frac{\sqrt{2} \kappa^5}{3 C_1}.$$

Note that Algorithm 1 maintains at most $h = O\left(\text{polylog}\left(T/\gamma\right)\right)$ parameters at each step $t$. Moreover, the matrix multiplications between $\tilde{W}_{t-1:t-m}$ and $\phi_i$ (in line 6 of Algorithm 1) for each time step $t \in [T]$, can be computed by zero-padding $\phi_i$ to be $T$-dimensional and performing online convolution with the disturbance stream $\{\mathbf{w}_t\}_{t \in [T]}$. Using the efficient online convolution technique introduced in Agarwal et al. (2024a) for this task, we obtain the following corollary:

**Corollary 2.2.** *The average running time of each of the $T$ executions of the inner loop in Algorithm 1 is $O\left(\log^4\left(T/\gamma\right)\right)$.*

**Proof Roadmap.** The proof of Theorem 2.1 follows a two-step strategy. First, we show that the class of diagonalizably stable linear policies can be approximated by a family of spectral controllers with bounded parameters (Lemma 4.3). Second, we analyze the regret of Algorithm 1, which performs projected online gradient descent over this spectral policy class. The analysis leverages convexity of the loss functions and boundedness of the feasible set. A high-level overview of the analysis appears in Section 4, and full proofs are provided in Sections A–D.

## 3 PRELIMINARIES

### 3.1 NOTATION

We use $\mathbf{x}$ to denote states, $\mathbf{u}$ for control inputs, and $\mathbf{w}$ for disturbances. The dimensions of the state and control spaces are denoted by $d = \dim(\mathbf{x})$ and $n = \dim(\mathbf{u})$, respectively. Matrices related to the system dynamics and control policy are denoted by capital letters $A, B, K, M$. For convenience, we write $\mathbf{w}_t = 0$ for all $t < 0$.

For any $t_2 \geq t_1$, we define $\tilde{W}_{t_2:t_1} \in \mathbb{R}^{d \times (t_2 - t_1 + 1)}$ as the matrix whose columns are $\mathbf{w}_{t_2}, \ldots, \mathbf{w}_{t_1}$, in that order. Additionally, given matrices $M_1, \ldots, M_h \in \mathbb{R}^{n \times d}$, we define $M_{1:h} \in \mathbb{R}^{n \times d \times h}$ as their concatenation along the third dimension.

Given a policy $\pi$, we denote the state and control at time $t$ by $(\mathbf{x}_t^\pi, \mathbf{u}_t^\pi)$ when following $\pi$. If $\pi$ is parameterized by a set of parameters $\Theta$, and the context makes the inputs clear, we use $(\mathbf{x}_t^\Theta, \mathbf{u}_t^\Theta)$ or $(\mathbf{x}_t(\Theta), \mathbf{u}_t(\Theta))$ to refer to the same quantities. For simplicity, we use $(\mathbf{x}_t, \mathbf{u}_t)$ without any superscript or argument to refer to the state and control at time $t$ under Algorithm 1.

### 3.2 SETTING

We now present the key definitions and outline the assumptions used throughout the paper. Our setting considers adversarial noise and convex cost functions. We begin by defining controllability:

**Definition 3.1.** *An LDS as in (1.1) is controllable if the noiseless LDS given by $\mathbf{x}_{t+1} = A\mathbf{x}_t + B\mathbf{u}_t$ can be steered to any target state from any initial state.*

Since the disturbances are non-stochastic, we assume without loss of generality that $\mathbf{x}_0 = 0$. The following assumptions formalize the notions of bounded disturbances and Lipschitz continuity of cost functions over bounded domains.

**Assumption 3.2.** *The system matrix $B$ is bounded, i.e., $\|B\| \leq \kappa_B$. The disturbance at each time step is also bounded, i.e., $\|\mathbf{w}_t\| \leq W$.*

**Assumption 3.3.** *The cost functions $c_t(\mathbf{x}, \mathbf{u})$ are convex. Moreover, as long as $\|\mathbf{x}\|, \|\mathbf{u}\| \leq D$, the gradients are bounded:*

$$\|\nabla_{\mathbf{x}} c_t(\mathbf{x}, \mathbf{u})\|, \|\nabla_{\mathbf{u}} c_t(\mathbf{x}, \mathbf{u})\| \leq GD.$$

In Definition 3.1 of Cohen et al. (2018), the notion of a $(\kappa, \gamma)$-strongly stable linear policy is introduced. Since our spectral analysis focuses on diagonalization, in Definition 3.4, we extend this to the notion of $(\kappa, \gamma)$-diagonalizably stable policies.:

**Definition 3.4.** *A linear policy $K$ is $(\kappa, \gamma)$-diagonalizably stable if there exist matrices $L, H$ satisfying $A + BK = HLH^{-1}$, such that the following conditions hold:*

1. *$L$ is diagonal with nonnegative entries.*

2. *The spectral norm of $L$ is strictly less than one, i.e., $\|L\| \leq 1 - \gamma$.*

3. *The controller and the transformation matrices are bounded, i.e., $\|K\|, \|H\|, \|H^{-1}\| \leq \kappa$.*

*We denote by $\mathcal{S} = \{K : K \text{ is } (\kappa, \gamma)\text{-diagonalizably stable}\}$ the set of such policies, and, with slight abuse of notation, also use $\mathcal{S}$ to refer to the class of linear policies $\mathbf{u}_t = S\mathbf{x}_t$ where $S \in \mathcal{S}$. Each policy in $\mathcal{S}$ is fully parameterized by the matrix $K \in \mathbb{R}^{n \times d}$.*

Definition 3.4 is similar to diagonal strong stability (Definition 2.3 of Agarwal et al. (2019b)), with the key difference that we require $L$ to have real eigenvalues.[1] Note that by Ackermann's formula (Ackermann, 1972; Galiaskarov et al., 2023), there always exists $K \in \mathcal{S}$ that controls the noiseless system. Finally, we introduce the following assumption for clarity of presentation, which we later relax in Section D.

**Assumption 3.5.** *The zero policy $K = 0$ lies in $\mathcal{S}$.*[2]

For simplicity, we assume that $\kappa, \kappa_B, W \geq 1$ and $\gamma \leq 2/3$, without loss of generality.

## 4 ANALYSIS OVERVIEW

We present the proof of our main result here, while deferring the proofs of technical lemmas to the appendix. Before proceeding, we outline the key technical considerations involved in establishing our final regret bound.

Algorithm 1 learns a convex relaxation of the policy class $\mathcal{S}$, referred to as the *spectral policy class*, defined as follows:

**Definition 4.1.** *[Spectral Controller] The class of Spectral Controllers with $h$ parameters, memory $m$ and stability $\gamma$ is defined as:*

$$\Pi_{h,m,\gamma}^{\mathsf{SC}} = \left\{ \pi_{h,m,\gamma,M}^{\mathsf{SC}}(\mathbf{w}_{t-1:t-m}) = \sum_{i=1}^{h} \sigma_i^{1/4} M_i \tilde{W}_{t-1:t-m} \phi_i \right\},$$

*where $\phi_i \in \mathbb{R}^m, \sigma_i \in \mathbb{R}$ are the $i^{\text{th}}$ top eigenvector and eigenvalue of $H \in \mathbb{R}^{m \times m}$ such that $H_{ij} = \frac{(1-\gamma)^{i+j-1}}{i+j-1}$. Any policy in this class is fully parameterized by the matrices $M_{1:h} \in \mathbb{R}^{n \times d \times h}$.*

To enable learning via online gradient descent, we require a bounded set of parameters:

**Definition 4.2.** *The set of bounded spectral parameters is defined as*

$$\mathcal{K} = \left\{ M_{1:h} \in \mathbb{R}^{h \times n \times d} \mid \left\| \mathbf{x}_t^M \right\|, \left\| \mathbf{u}_t^M \right\| \leq \frac{3\kappa^3 W}{\gamma}, \|M_{1:h}\| \leq \kappa^3 \sqrt{\frac{2h}{\gamma}} \right\}.$$

In Section A, we prove that the spectral policy class can approximate $\mathcal{S}$ up to arbitrary accuracy. Formally, we state this as:

---

[1]The requirement of nonnegative eigenvalues can be relaxed by integrating over a larger set; it is imposed here for ease of presentation.

[2]Assumption 3.5 can be relaxed via a simple reduction, as outlined in Section D. Theorem 2.1 holds without this assumption, up to an additional factor of $\kappa$ in $C_1$.

**Lemma 4.3.** *For any linear policy $K \in \mathcal{S}$, there exists a SC policy with $M \in \mathcal{K}$ such that for any $\varepsilon \in (0,1)$:*

$$\sum_{t=1}^{T} \left| c_t(\mathbf{x}_t^M, \mathbf{u}_t^M) - c_t(\mathbf{x}_t^K, \mathbf{u}_t^K) \right| \le \varepsilon T,$$

*if (i) $m = \left\lceil \frac{1}{\gamma} \log \left( \frac{8C_1}{\varepsilon \gamma^3} \right) \right\rceil$ and (ii) $h \ge 2 \log T \log \left( \frac{900 C_1 d}{\varepsilon \gamma^3} \log T \log^{1/4} \left( \frac{2}{\gamma} \right) \log^{1/2} \left( \frac{8C_1}{\varepsilon \gamma^3} \right) \right)$ where $C_1$ is as defined in Theorem 2.1.*

We further note that in Algorithm 1, online gradient descent is not performed on the actual cost function, but on a modified cost function, referred to as the memory-less loss function:

**Definition 4.4.** *We define the memory-less loss function at time $t$ as*

$$\ell_t(M_{1:h}) = c_t(\mathbf{x}_t(M_{1:h}), \mathbf{u}_t(M_{1:h})). \tag{4.1}$$

Classical results in online gradient descent provide a regret bound with respect to loss functions $\ell_t(M_{1:h}^t)$. However, our regret is defined in terms of the actual costs $c_t(\mathbf{x}_t, \mathbf{u}_t)$. Nevertheless, in Section B.3, we prove that $c_t(\mathbf{x}_t, \mathbf{u}_t)$ is well approximated by $\ell_t(M_{1:h}^t)$. We formally state this result as:

**Lemma 4.5.** *Algorithm 1 is executed with $\eta = C_2 \sqrt{\frac{\gamma^3}{Tmh}}$. Then for every $t \in [T]$,*

$$\left| c_t(\mathbf{x}_t, \mathbf{u}_t) - \ell_t(M_{1:h}^t) \right| \le \frac{6C_1 \sqrt{mh}}{\gamma^{7/2} \sqrt{T}} \log^{1/4} \left( \frac{2}{\gamma} \right),$$

*where $C_1$ and $C_2$ are as defined in Theorem 2.1.*

*Proof of Theorem 2.1.* For $\varepsilon = 1/\sqrt{T}$, observe that our choice of $m$ and $h$ satisfies the conditions in Lemma 4.3 (using the fact that $T, d, C \ge 1$ and $0 < \gamma < 1$). Hence, using Lemma 4.3 and the Definition 4.4 we get:

$$\min_{M^\star \in \mathcal{K}} \sum_{t=1}^{T} \ell_t(M_{1:h}^\star) - \min_{K \in \mathcal{S}} \sum_{t=1}^{T} c_t(\mathbf{x}_t^K, \mathbf{u}_t^K) \le \sqrt{T} \le \frac{C_1 \sqrt{T}}{\gamma^4} \log^3 \left( \frac{C_1 T d}{\gamma^3} \right). \tag{4.2}$$

We now invoke the regret of the Online Gradient Descent, stated in Theorem 3.1 in Hazan (2016). By Lemma B.1 the set $\mathcal{K}$ is convex, and by Lemma B.2 the memory-less loss functions are convex functions. Furthermore, using the bound on lipschitz constant of the memory-less loss computed in Lemma B.3 and the bound of the diameter of $\mathcal{K}$, for our choice of $\eta$, we get a regret bound. For our choice of $h$ and $m$, the regret bound evaluates to:

$$\sum_{t=1}^{T} \ell_t \left( M_{1:h}^t \right) - \min_{M^\star \in \mathcal{K}} \sum_{t=1}^{T} \ell_t \left( M_{1:h}^\star \right) \le \frac{384 C_1 \sqrt{T}}{\gamma^4} \log^3 \left( \frac{900 C_1 d T}{\gamma^3} \right). \tag{4.3}$$

Next, Lemma 4.5 gives us the following bound on the difference between the memory-less loss $\ell_t(M_{1:h}^t)$ and the cost incurred by the algorithm $c_t(\mathbf{x}_t, \mathbf{u}_t)$ for every $t \in [T]$. Taking the sum over all $t \in [T]$:

$$\sum_{t=1}^{T} c_t \left( \mathbf{x}_t, \mathbf{u}_t \right) - \sum_{t=1}^{T} \ell_t \left( M_{1:h}^t \right) \le \frac{24 C_1 \sqrt{T}}{\gamma^4} \log^3 \left( \frac{900 C_1 d T}{\gamma^3} \right). \tag{4.4}$$

Finally, we add equations (4.2),(4.3),(4.4) together to obtain the result. $\qquad \square$

## 5 EXPERIMENTS

We compare our method to GPC. Unless otherwise noted, we control an LDS with state dimension $d = 10$ and a single control input $n = 1$. Disturbances are Gaussian and the cost is a fixed quadratic function, randomly generated at $t = 0$ and normalized by the dimensions. The system matrices are sampled as follows: $A$ is diagonal with entries drawn uniformly from $[0.5, 0.95]$, and $B$ has

entries i.i.d. $\mathcal{N}(0, 1)$. Training uses Adam Kingma & Ba (2014) with learning rate $10^{-4}$, $\beta_1 = 0.9$, $\beta_2 = 0.999$, and projection onto the unit ball at each step. Performance is averaged over 20 runs, and we report the mean of the last 100 observed costs at each time step. We report loss (rather than regret), since the optimal comparator in hindsight is not directly observable for a realized disturbance sequence. Both methods use the last 50 disturbances as input features; GPC therefore has $50 \times d$ parameters, while OSC uses only the top 15 Hankel eigenvectors, totaling $15 \times d$ parameters.

## 5.1 COMPARISON OF DIFFERENT MODELS AND SYSTEMS

One way to view GPC is as performing linear regression on past disturbances. Analogously, Algorithm 1 can be interpreted as linear regression on the convolution of disturbances with spectral filters. From this perspective, the method amounts to linear regression on a more expressive set of features: the spectral filters compress the information in the disturbances while preserving the structure most relevant for prediction. While we have a solid theoretical understanding in the linear setting, modern deep learning methods often benefit from richer models. Motivated by this, we replace the linear model with a neural network operating on the same spectral features, and in this section we demonstrate the advantages that these features provide.

We evaluate GPC and OSC on both linear and nonlinear systems, and with either a simple linear regression model or an two-layer ReLU neural network, with 100 hidden units. The nonlinear system (LDS ReLU) follows

$$x_{t+1} = \text{ReLU}(Ax_t + Bu_t) + w_t.$$

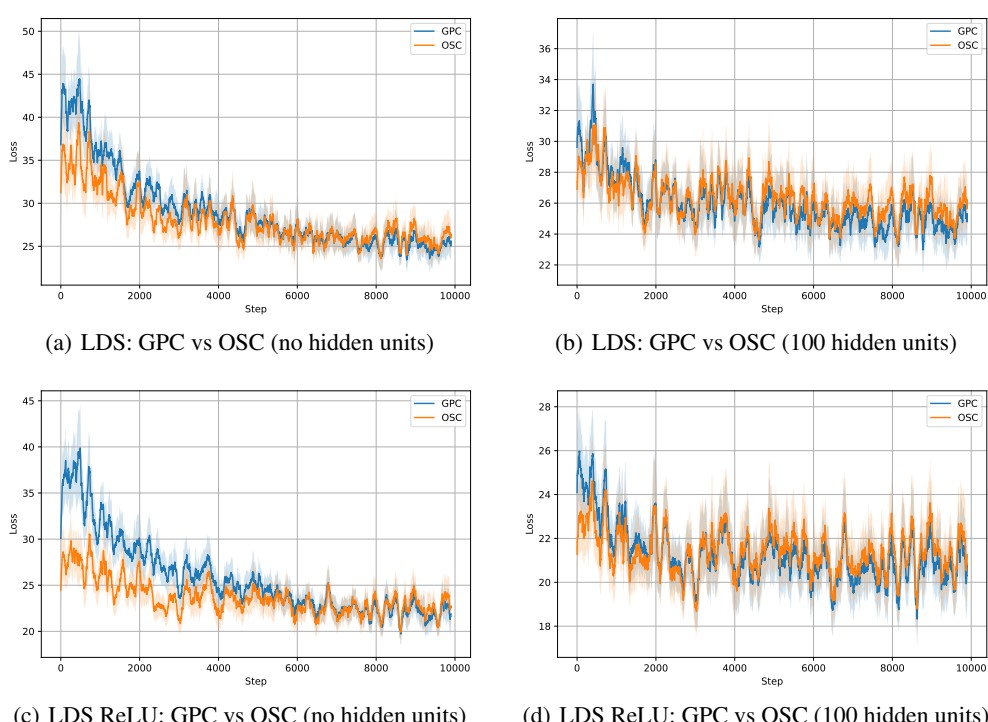

(a) LDS: GPC vs OSC (no hidden units)

(b) LDS: GPC vs OSC (100 hidden units)

(c) LDS ReLU: GPC vs OSC (no hidden units)

(d) LDS ReLU: GPC vs OSC (100 hidden units)

Figure 2: Comparison of GPC and OSC under linear (LDS) and nonlinear (LDS ReLU) dynamics, using either a linear head (no hidden units) or a 100-unit MLP head.

Across settings, OSC uses fewer parameters yet achieves long-run performance comparable to GPC. Hidden layers improve both methods, showing that spectral features remain effective with nonlinear models and can serve as inputs to more powerful architectures (e.g., CNNs or Transformers), potentially amplifying their empirical benefits.

## 5.2 ABLATING STATE DIMENSIONALITY

We study the effect of the state dimension $d$ with control input fixed at $n = 1$. Figure 3 reports results for $d \in \{2, 5, 10, 20\}$, with loss normalized by $d$. Performance degrades as $d$ grows, consistent with the intrinsic difficulty of controlling larger state spaces through a single input channel.

## 5.3 ABLATING NUMBER OF PARAMETERS IN OSC

OSC uses far fewer parameters than GPC: with memory 50, GPC requires $50d$ parameters, whereas OSC uses only $hd$, where $h$ is the number of Hankel eigenvectors. Figure 4 shows that performance improves with $h$ up to about $h = 20$, beyond which returns diminish. Even with $h = 5$, OSC is competitive with GPC, and by $h = 20$ their performance is nearly identical, highlighting the compactness of spectral features.

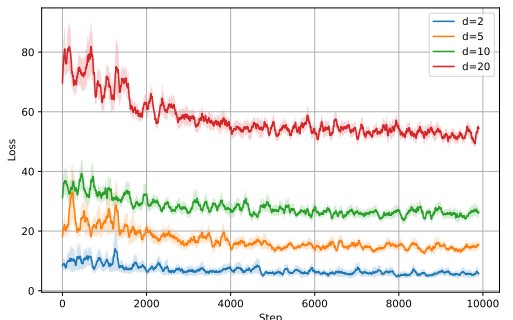
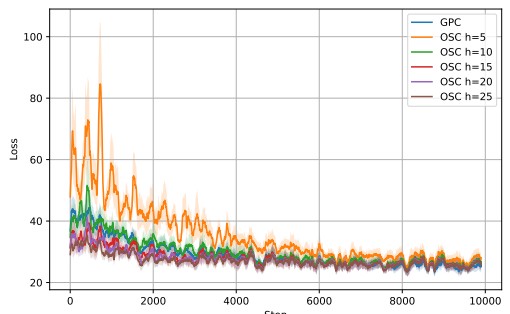

Figure 3: Effect of state dimensionality (loss normalized by $d$): higher dimensions make it harder to control an LDS with a single input.

Figure 4: Effect of spectral parameters $h$: even small $h$ gives strong performance, and by $h = 20$ OSC matches GPC with far fewer parameters.

## 5.4 ABLATING STABILITY

We now examine how system stability affects performance. We vary the distribution of the diagonal entries of $A$: previously drawn uniformly from $[0.5, 0.95]$, and now from $[0.5, 0.99]$ and $[0.5, 0.999]$. Larger eigenvalues correspond to a smaller stability margin, making the control problem harder. Figure 5 shows that while overall loss increases in the less stable regime, OSC and GPC maintain comparable performance.

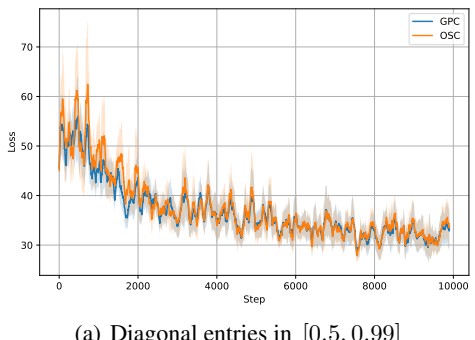
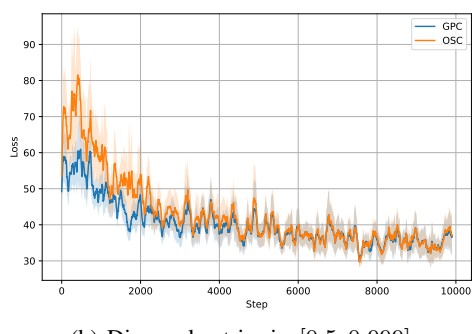

(a) Diagonal entries in $[0.5, 0.99]$

(b) Diagonal entries in $[0.5, 0.999]$

Figure 5: Effect of system stability on performance. Smaller stability margins (larger eigenvalues of $A$) make the problem harder, but OSC remains competitive with GPC across regimes.

## 5.5 ABLATING DISTURBANCES

We next study the effect of the disturbance distribution. Beyond Gaussian noise, we consider Rademacher noise (i.i.d. $\pm 1$) and a deterministic sinusoid. Figure 6 shows that OSC and GPC behave similarly across disturbance types. When disturbances follow a simple nonlinear rule (e.g., sinusoid), spectral filtering is less natural than directly learning the rule, but OSC still remains competitive.

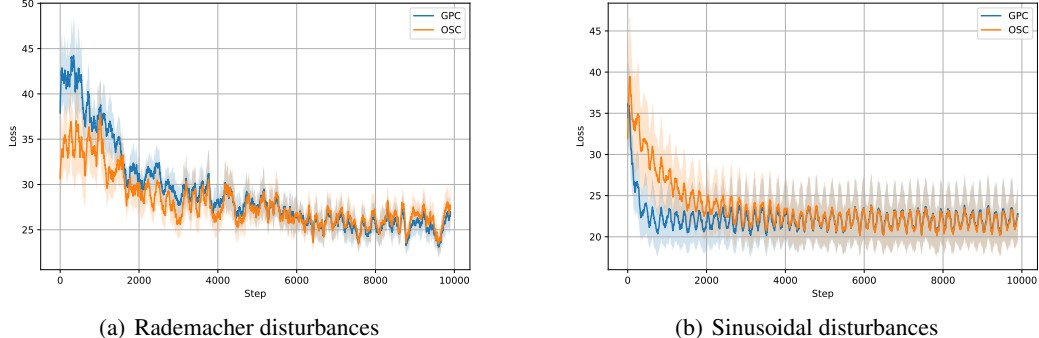

(a) Rademacher disturbances          (b) Sinusoidal disturbances

Figure 6: Effect of disturbance distribution on performance. OSC and GPC exhibit similar relative behavior under random (Rademacher) and structured (sinusoidal) disturbances.

## 6 CONCLUSION AND DISCUSSION

We introduced a spectral-filtering approach for online control of linear dynamical systems with adversarial disturbances. By relaxing linear policies into Hankel-based spectral features, we obtain an efficient convex formulation that preserves optimal regret while greatly reducing runtime and parameters. These spectral features provide compact representations of dynamics and open the door to integration with modern deep learning models for large-scale control.

## ACKNOWLEDGMENTS

The authors thank Karan Singh and Zhou Lou for their valuable comments. EH gratefully acknowledges support from the Office of Naval Research, and Open Philanthropy.

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

# A APPROXIMATION RESULTS

To prove Lemma 4.3, we shall show that the class of $(\kappa, \gamma)$-diagonalizably stable linear policies can be approximated using Open-Loop Optimal Controllers (Definition A.1) in Lemma A.2. We then show that the class of open-loop optimal controllers can be approximated by the class of spectral controllers in Lemma A.3. We begin by defining the class of open-loop optimal controllers as follows:

**Definition A.1** (Open Loop Optimal Controller). *The class of Open Loop Optimal Controllers of with memory $m$ is defined as:*

$$\Pi_m^{\mathsf{OLOC}} = \left\{ \pi_{m,K}^{\mathsf{OLOC}}(\mathbf{w}_{t-1:t-m}) = K \sum_{i=1}^{m} (A + BK)^{i-1} \mathbf{w}_{t-i} \right\} .$$

*Any policy in this class is fully parameterized by the matrix $K \in \mathbb{R}^{d \times n}$ and the memory $m \in \mathbb{Z}$.*

Next, we state and prove Lemma A.2, which shows that any linear policy in $\mathcal{S}$ can be approximated up to arbitrary accuracy with an open-loop optimal controller of suitable memory.

**Lemma A.2.** *Let a linear policy $K \in \mathcal{S}$. Then, for $m \geq \frac{1}{\gamma} \log \left( \frac{8 G \kappa_B \kappa^8 W^2}{\varepsilon \gamma^3} \right)$ and $\varepsilon \in (0, 1)$,*

$$\sum_{t=1}^{T} \left| c_t(\mathbf{x}_t^{K,m}, \mathbf{u}_t^{K,m}) - c_t(\mathbf{x}_t^K, \mathbf{u}_t^K) \right| \leq \frac{\varepsilon}{2} T , \qquad \left\| \mathbf{x}_t^{K,m} \right\|, \left\| \mathbf{u}_t^{K,m} \right\| \leq \frac{2 \kappa^3 W}{\gamma} .$$

*Proof.* We begin by bounding the difference in the state and the difference in the control inputs. The difference in cost is bounded using the fact that the cost functions are lipschitz in the control and state. We begin by unrolling the expressions for $\mathbf{u}_t^K$ and $\mathbf{x}_t^K$ in terms of $\mathbf{w}_t$:

$$\mathbf{u}_t^K = K \sum_{i=1}^{t} (A + BK)^{i-1} \mathbf{w}_{t-i} , \qquad \mathbf{x}_t^K = \sum_{i=1}^{t} A^{i-1} \mathbf{w}_{t-i} + \sum_{i=1}^{t} A^{i-1} B \mathbf{u}_{t-i}^K . \qquad \text{(A.1)}$$

Notice that for all $t \leq m$, $\mathbf{u}_t^K = \mathbf{u}_t^{K,m}$ and hence $\mathbf{x}_t^K = \mathbf{x}_t^{K,m}$. For $t > m$,

$$\begin{aligned} \left\| \mathbf{u}_t^K - \mathbf{u}_t^{K,m} \right\| &= \left\| K \sum_{i=1}^{t} (A + BK)^{i-1} \mathbf{w}_{t-i} - K \sum_{i=1}^{m} (A + BK)^{i-1} \mathbf{w}_{t-i} \right\| \\ &= \left\| K \sum_{i=m+1}^{t} (A + BK)^{i-1} \mathbf{w}_{t-i} \right\| \\ &\leq \kappa^3 W \sum_{i=m+1}^{t} (1 - \gamma)^{i-1} \qquad\qquad [\Delta - \text{ineq., C-S}] \\ &\leq \frac{\kappa^3 W}{\gamma} (1 - \gamma)^m . \end{aligned}$$

Using this together with (A.1) and the fact that $\mathbf{u}_t^K = \mathbf{u}_t^{K,m}$ for any $t \leq m$, we similarly get:

$$\left\| \mathbf{x}_t^K - \mathbf{x}_t^{K,m} \right\| = \left\| \sum_{i=1}^{t} A^{i-1} B (\mathbf{u}_{t-i}^K - \mathbf{u}_{t-i}^{K,m}) \right\| = \left\| \sum_{i=1}^{t-m} A^{i-1} B (\mathbf{u}_{t-i}^K - \mathbf{u}_{t-i}^{K,m}) \right\| ,$$

and by using Assumption 3.5 we can write:

$$\left\| \mathbf{x}_t^K - \mathbf{x}_t^{K,m} \right\| \leq \kappa_B \kappa^2 \sum_{i=1}^{\infty} (1 - \gamma)^{i-1} \left\| \mathbf{u}_{t-i}^K - \mathbf{u}_{t-i}^{K,m} \right\| \leq \frac{\kappa_B \kappa^5 W}{\gamma^2} (1 - \gamma)^m .$$

Using the fact that $\kappa_B \kappa^2 / \gamma > 1$, we get the following uniform bound:

$$\max \left\{ \left\| \mathbf{u}_t^K - \mathbf{u}_t^{K,m} \right\|, \left\| \mathbf{x}_t^K - \mathbf{x}_t^{K,m} \right\| \right\} \leq \frac{\kappa_B \kappa^5 W}{\gamma^2} (1 - \gamma)^m .$$

Whenever $m \geq \frac{1}{\gamma} \log \left( \frac{\kappa_B \kappa^2}{\gamma} \right)$, which is indeed true from our choice of $\varepsilon$ and $m$, this implies that the $\left\| \mathbf{x}_t^K - \mathbf{x}_t^{K,m} \right\|, \left\| \mathbf{u}_t^K - \mathbf{u}_t^{K,m} \right\| \leq \kappa^3 W / \gamma$. Using Lemma A.4 $\left\| \mathbf{x}_t^K \right\|, \left\| \mathbf{u}_t^K \right\| \leq \kappa^3 W / \gamma$, by triangle inequality $\left\| \mathbf{x}_t^{K,m} \right\|, \left\| \mathbf{u}_t^{K,m} \right\| \leq 2\kappa^3 W / \gamma$. Thus, the sum of costs is bounded using lipschitzness of $c_t$ as follows:

$$\sum_{t=1}^{T} \left| c_t(\mathbf{x}_t^K, \mathbf{u}_t^K) - c_t(\mathbf{x}_t^{K,m}, \mathbf{u}_t^{K,m}) \right| \leq \frac{2G\kappa^3 W}{\gamma} \sum_{t=1}^{T} \left( \left\| \mathbf{x}_t^K - \mathbf{x}_t^{K,m} \right\| + \left\| \mathbf{u}_t^K - \mathbf{u}_t^{K,m} \right\| \right)$$

$$\leq \frac{4G\kappa_B \kappa^8 W^2}{\gamma^3} (1 - \gamma)^m \cdot T \leq \frac{\varepsilon}{2} T. \qquad \text{[choice of } m]$$

$\square$

We shall now prove that every open-loop optimal controller can be approximated up to arbitrary accuracy with a spectral controller.

**Lemma A.3.** *For every open loop optimal controller* $\pi_{K,m}^{\mathsf{OLOC}}$ *such that* $K \in \mathcal{S}$ *and* $\left\| \mathbf{x}^{K,m} \right\|, \left\| \mathbf{u}^{K,m} \right\| \leq 2\kappa^3 W / \gamma$, *there exists a spectral controller* $\pi_{h,m,\gamma,M}^{\mathsf{SC}}$ *with* $M \in \mathcal{K}$ *such that:*

$$\sum_{t=1}^{T} \left| c_t(\mathbf{x}_t^M, \mathbf{u}_t^M) - c_t(\mathbf{x}_t^{K,m}, \mathbf{u}_t^{K,m}) \right| \leq \frac{\varepsilon}{2} T.$$

*for any* $\varepsilon \in (0,1)$ *and* $h \geq 2 \log T \log \left( \frac{600 G \kappa_B \kappa^8 W^2 \sqrt{m} d}{\varepsilon \gamma^{5/2}} \log T \log^{1/4} \left( \frac{2}{\gamma} \right) \right)$.

*Proof.* Since $K \in \mathcal{S}$, there exists a diagonal $L \in \mathbb{R}^{n \times n}$ as in Definition 3.4 so that:

$$\mathbf{u}_t^{K,m} = K \sum_{i=1}^{m} (A + BK)^{i-1} \mathbf{w}_{t-i} = K \sum_{i=1}^{m} H L^{i-1} H^{-1} \mathbf{w}_{t-i}.$$

Then write $L^{i-1} = \sum_{j=1}^{d} \alpha_j^{i-1} e_j e_j^\top$ and obtain

$$\mathbf{u}_t^{K,m} = K \sum_{i=1}^{m} H \left( \sum_{j=1}^{d} \alpha_j^{i-1} e_j e_j^\top \right) H^{-1} \mathbf{w}_{t-i} = K \sum_{j=1}^{d} H e_j e_j^\top H^{-1} \sum_{i=1}^{m} \alpha_j^{i-1} \mathbf{w}_{t-i}.$$

Recall $\tilde{W}_{t-1:t-m} = [\mathbf{w}_{t-1}, \ldots, \mathbf{w}_{t-m}] \in \mathbb{R}^{d \times m}$, define $\boldsymbol{\mu}_\alpha = [1, \alpha, \ldots, \alpha^{m-1}] \in \mathbb{R}^m$ and get

$$\mathbf{u}_t^{K,m} = K \sum_{j=1}^{d} H e_j e_j^\top H^{-1} \tilde{W}_{t-1:t-m} \boldsymbol{\mu}_{\alpha_j}$$

$$= K \sum_{j=1}^{d} H e_j e_j^\top H^{-1} \tilde{W}_{t-1:t-m} \left( \sum_{i=1}^{m} \boldsymbol{\phi}_i \boldsymbol{\phi}_i^\top \right) \boldsymbol{\mu}_{\alpha_j} \qquad \left[ \sum_{i=1}^{m} \boldsymbol{\phi}_i \boldsymbol{\phi}_i^\top = \mathbb{I}_m \right]$$

$$= K \sum_{i=1}^{m} \left( \sum_{j=1}^{d} H e_j e_j^\top H^{-1} \boldsymbol{\phi}_i^\top \boldsymbol{\mu}_{\alpha_j} \right) \tilde{W}_{t-1:t-m} \boldsymbol{\phi}_i.$$

Let $\pi_{h,m,\gamma,M^*}^{\mathsf{SC}}$ be the spectral controller with $M_i^* = \sigma_i^{-1/4} K H \left( \sum_{j=1}^{d} \boldsymbol{\phi}_i^\top \boldsymbol{\mu}_{\alpha_j} e_j e_j^\top \right) H^{-1}$ for all $i \in [h]$. Note that we have

$$\| M_i^* \| \leq \kappa^3 \cdot \max_{\ell \in [d]} \sigma_j^{-1/4} \langle \boldsymbol{\phi}_j, \boldsymbol{\mu}_{\alpha_l} \rangle \quad \forall 1 \leq j \leq m,$$

and from the analysis of Lemma C.4, we have that $\sigma_j^{-1/4}\langle\phi_j, \mu(\alpha_l)\rangle \leq \sqrt{\frac{2}{\gamma}}$. Thus, $\|M_{1:h}^*\| \leq \kappa^3\sqrt{\frac{2h}{\gamma}}$. Then,

$$
\begin{aligned}
\left\|\mathbf{u}_t^{K,m} - \mathbf{u}_t^{M^*}\right\| &= \left\|K\sum_{i=h+1}^{m} H\left(\sum_{j=1}^{d}\phi_i^\top\boldsymbol{\mu}_{\alpha_j}e_j e_j^\top\right)H^{-1}\tilde{W}_{t-1:t-m}\phi_i\right\| \\
&\leq \kappa^3 W\sqrt{m}\sum_{i=h+1}^{m}\sum_{j=1}^{d}|\phi_i^\top\boldsymbol{\mu}_{\alpha_j}| \qquad\qquad \left[\left\|\tilde{W}_{t-1:t-m}\right\| \leq W\sqrt{m}\right] \\
&\leq \frac{30\kappa^3 W\sqrt{m}}{\sqrt{\gamma}}\log^{1/4}\left(\frac{2}{\gamma}\right)\sum_{i=h+1}^{m}\sum_{j=1}^{d}\exp\left(-\frac{\pi^2 j}{16\log T}\right) \qquad [\text{Lemma C.4}] \\
&\leq \frac{30\kappa^3 W\sqrt{m}d}{\sqrt{\gamma}}\log^{1/4}\left(\frac{2}{\gamma}\right)\int_h^\infty \exp\left(-\frac{\pi^2 j}{16\log T}\right)dx \\
&\leq \frac{50\kappa^3 W\sqrt{m}d}{\sqrt{\gamma}}\log T\log^{1/4}\left(\frac{2}{\gamma}\right)\exp\left(-\frac{\pi^2 h}{16\log T}\right),
\end{aligned}
$$

$$
\begin{aligned}
\left\|\mathbf{x}_t^{M^*} - \mathbf{x}_t^{K,m}\right\| &= \left\|\sum_{i=1}^{t}A^{i-1}B(\mathbf{u}_{t-i}^{M^*} - \mathbf{u}_{t-i}^{K,m})\right\| \\
&\leq \kappa_B\kappa^2\sum_{i=1}^{t}(1-\gamma)^{i-1}\left\|\mathbf{u}_{t-i}^{M} - \mathbf{u}_{t-i}^{K,m}\right\| \qquad [\text{Assumption 3.5}] \\
&\leq \frac{50\kappa_B\kappa^5 W\sqrt{m}d}{\gamma^{3/2}}\log T\log^{1/4}\left(\frac{2}{\gamma}\right)\exp\left(-\frac{\pi^2 h}{16\log T}\right).
\end{aligned}
$$

Using the fact that $\kappa_B\kappa^2/\gamma > 1$, we get a uniform bound:

$$
\max\left\{\left\|\mathbf{x}_t^{K,m} - \mathbf{x}_t^{M^*}\right\|, \left\|\mathbf{u}_t^{K,m} - \mathbf{u}_t^{M^*}\right\|\right\} \leq \frac{50\kappa_B\kappa^5 W\sqrt{m}d}{\gamma^{3/2}}\log T\log^{1/4}\left(\frac{2}{\gamma}\right)\exp\left(-\frac{\pi^2 h}{16\log T}\right).
$$

Whenever $h \geq 2\log T\log\left(\frac{50\kappa_B\kappa^2\sqrt{m}d}{\sqrt{\gamma}}\log T\log\left(\frac{2}{\gamma}\right)\right)$, which is indeed the case for our choice of $\varepsilon$ and $h$, this implies that $\left\|\mathbf{x}_t^{M^*} - \mathbf{x}_t^{K,m}\right\|, \left\|\mathbf{u}_t^{M^*} - \mathbf{u}_t^{K,m}\right\| \leq \kappa^3 W/\gamma$. Hence, by triangle inequality $\left\|\mathbf{x}_t^{M^*}\right\|, \left\|\mathbf{u}_t^{M^*}\right\| \leq 3\kappa^2 W/\gamma$. Thus, the sum of costs is bounded as follows:

$$
\begin{aligned}
\sum_{t=1}^{T}\left|c_t(\mathbf{x}_t^{M^*}, \mathbf{u}_t^{M^*}) - c_t(\mathbf{x}_t^{K,m}, \mathbf{u}_t^{K,m})\right| &\leq \frac{3G\kappa^3 W}{\gamma}\sum_{t=1}^{T}\left(\left\|\mathbf{x}_t^{M^*} - \mathbf{x}_t^{K,m}\right\| + \left\|\mathbf{u}_t^{M^*} - \mathbf{u}_t^{K,m}\right\|\right) \\
&\leq \frac{300G\kappa_B\kappa^8 W^2\sqrt{m}d}{\gamma^{5/2}}\log T\log^{1/4}\left(\frac{2}{\gamma}\right)\exp\left(-\frac{\pi^2 h}{16\log T}\right)\cdot T \\
&\leq \frac{\varepsilon}{2}T. \qquad\qquad [\text{choice of } h]
\end{aligned}
$$

$\square$

We conclude the proof of Lemma 4.3 using Lemmas A.2 and A.3 as follows:

*Proof of Lemma 4.3*: $K$ is a $(\kappa, \gamma)-$diagonalizably stable linear policy and $\varepsilon \in (0,1)$. Hence, by Lemma $A.2$, for the choice of $m = \left\lceil\frac{1}{\gamma}\log\left(\frac{8G\kappa_B\kappa^8 W^2}{\varepsilon\gamma^3}\right)\right\rceil$, we have that

$$
\sum_{t=1}^{T}\left|c_t(\mathbf{x}_t^{K,m}, \mathbf{u}_t^{K,m}) - c_t(\mathbf{x}_t^{K}, \mathbf{u}_t^{K})\right| \leq \frac{\varepsilon}{2}T, \qquad\qquad \left\|\mathbf{x}_t^{K,m}\right\|, \left\|\mathbf{u}_t^{K,m}\right\| \leq \frac{2\kappa^3 W}{\gamma}.
$$

Since $\left\|\mathbf{x}_t^{K,m}\right\|, \left\|\mathbf{u}_t^{K,m}\right\| \leq \frac{2\kappa^3 W}{\gamma}$, for any $h \geq 2\log T \log\left(\frac{600 G\kappa_B \kappa^8 W^2 \sqrt{m}d}{\varepsilon\gamma^{5/2}}\log T \log^{1/4}\left(\frac{2}{\gamma}\right)\right)$, there exists an $M \in \mathcal{K}$ such that:

$$\sum_{t=1}^{T}\left|c_t(\mathbf{x}_t^M, \mathbf{u}_t^M) - c_t(\mathbf{x}_t^{K,m}, \mathbf{u}_t^{K,m})\right| \leq \frac{\varepsilon}{2}T.$$

In particular, since $2x \geq \lceil x \rceil$ for $x > 0.5$ and since $\frac{1}{\gamma}\log\left(\frac{8 G\kappa_B \kappa^8 W^2}{\varepsilon\gamma^3}\right) \geq \log(8) > 0.5$, $m \leq \frac{2}{\gamma}\log\left(\frac{8 G\kappa_B \kappa^8 W^2}{\varepsilon\gamma^3}\right)$. Thus, for $h \geq 2\log T \log\left(\frac{900 G\kappa_B \kappa^8 W^2 d}{\varepsilon\gamma^3}\log T \log^{1/4}\left(\frac{2}{\gamma}\right)\log^{1/2}\left(\frac{8 G\kappa_B \kappa^8 W^2}{\varepsilon\gamma^3}\right)\right)$ there exists such an $M \in \mathcal{K}$. Using triangle inequality, we get:

$$\sum_{t=1}^{T}\left|c_t(\mathbf{x}_t^M, \mathbf{u}_t^M) - c_t(\mathbf{x}_t^K, \mathbf{u}_t^K)\right| \leq \varepsilon T.$$

$\square$

Finally, we derive the bound on the state and control obtained by following a linear policy from $\mathcal{S}$, which we use in the proof of Lemma A.2.

**Lemma A.4.** *For any $K \in \mathcal{S}$, the corresponding states $\mathbf{x}_t^K$ and control inputs $\mathbf{u}_t^K$ are bounded by*

$$\left\|\mathbf{x}_t^K\right\| \leq \frac{\kappa^2 W}{\gamma}, \quad \left\|\mathbf{u}_t^K\right\| \leq \frac{\kappa^3 W}{\gamma}.$$

*Proof.* As in many other parts of this paper, we first write the states as a linear transformation of the disturbances:

$$\begin{aligned}\left\|\mathbf{x}_t^K\right\| &= \left\|\sum_{i=1}^{t}(A+BK)^{i-1}\mathbf{w}_{t-i}\right\| \\ &\leq \sum_{i=0}^{t}(1-\gamma)^i\left\|\mathbf{w}_{t-i-1}\right\| \\ &\leq \frac{\kappa^2 W}{\gamma}.\end{aligned}$$

Then, since $\mathbf{u}_t^K = K\mathbf{x}_t^K$ with $\|K\| \leq \kappa$, we obtain the result. $\square$

# B LEARNING RESULTS

## B.1 CONVEXITY OF LOSS FUNCTION AND FEASIBILITY SET

To conclude the analysis, we first show that the feasibility set $\mathcal{K}$ is convex and the loss functions are convex with respect to the variables $M_{1:h}$. This follows since the states and the controls are linear transformations of the variables.

**Lemma B.1.** *The set $\mathcal{K} = \left\{M_{1:h} \in \mathbb{R}^{h \times n \times d} \mid \left\|\mathbf{x}_t^M\right\|, \left\|\mathbf{u}_t^M\right\| \leq \frac{3\kappa^3 W}{\gamma}, \|M_{1:h}\| \leq \kappa^3\sqrt{\frac{2h}{\gamma}}\right\}$ is convex.*

*Proof.* Since $\mathbf{x}_t^M, \mathbf{u}_t^M$ are linear in $M_{1:h}$, from the convexity of the norm, the fact that the sublevel sets of a convex function is convex and that the intersection of convex sets is convex, we are done. $\square$

**Lemma B.2.** *The loss $\ell_t(M_{1:h})$ is convex in $M_{1:h}$.*

*Proof.* The loss function $\ell_t$ is given by $\ell_t(M_{1:h}) = c_t(\mathbf{x}_t(M_{1:h}), \mathbf{u}_t(M_{1:h}))$. Since the cost $c_t$ is a convex function with respect to its arguments, we simply need to show that $\mathbf{x}_t^M$ and $\mathbf{u}_t^M$ depend linearly on $M_{1:h}$. The state is given by

$$\mathbf{x}_t^M = A\mathbf{x}_{t-1}^M + B\mathbf{u}_{t-1}^M + \mathbf{w}_{t-1} = A\mathbf{x}_{t-1}^M + B\left(\sum_{i=1}^{h} \sigma_i^{1/4} M_i \tilde{W}_{t-2:t-m-1}\phi_i\right) + \mathbf{w}_{t-1}\,.$$

By induction, we can further simplify

$$\mathbf{x}_t^M = \sum_{i=1}^{t} A^{i-1}\mathbf{w}_{t-i} + \sum_{i=1}^{t} A^{i-1}B \sum_{j=1}^{h} \sigma_j^{1/4} M_j \tilde{W}_{t-i-1:t-i-m}\phi_j\,,$$

which is a linear function of the variables. Similarly, the control $\mathbf{u}_t$ is given by

$$\mathbf{u}_t^M = \sum_{i=1}^{h} \sigma_i^{1/4} M_i \tilde{W}_{t-1:t-m}\phi_i\,.$$

Thus, we have shown that $\mathbf{x}_t(M_{1:h})$ and $\mathbf{u}_t(M_{1:h})$ are linear transformations of $M_{1:h}$. A composition of convex and linear functions is convex, which concludes our Lemma. $\qquad\square$

## B.2  Lipschitzness of $\ell_t(\cdot)$

The following lemma states and proves the explicit lipschitz constant of $\ell_t(\cdot)$.

**Lemma B.3.** *For any $M_{1:h}, M'_{1:h} \in \mathcal{K}$ it holds that,*

$$|\ell_t(M_{1:h}) - \ell_t(M'_{1:h})| \leq \frac{6G\kappa_B\kappa^5 W^2\sqrt{m}h}{\gamma^2}\log^{1/4}\left(\frac{2}{\gamma}\right)\|M_{1:h} - M'_{1:h}\|\,.$$

*Proof.* Taking the difference in controls:

$$\|\mathbf{u}_t(M_{1:h}) - \mathbf{u}_t(M'_{1:h})\| = \left\|\sum_{j=1}^{h} \sigma_j^{1/4}(M_j - M'_j)\tilde{W}_{t-1:t-m}\phi_j\right\|$$

$$\leq W\sqrt{m}\log^{1/4}\left(\frac{2}{\gamma}\right)\sum_{j=1}^{h}\|M_j - M'_j\|\,.$$

By unrolling the recursion, we have:

$$\mathbf{x}_t(M_{1:h}) = \sum_{i=1}^{t} A^{i-1}\mathbf{w}_{t-i} + \sum_{i=1}^{t} A^{i-1}B \sum_{j=1}^{h} \sigma_j^{1/4} M_j \tilde{W}_{t-i-1:t-i-m}\phi_j\,,$$

Taking the difference,

$$\|\mathbf{x}_t(M_{1:h}) - \mathbf{x}_t(M'_{1:h})\| = \left\|\sum_{i=1}^{t} A^{i-1}B \sum_{j=1}^{h} \sigma_j^{1/4}(M_j - M'_j)\tilde{W}_{t-i-1:t-i-m}\phi_j\right\|$$

$$\leq \sum_{i=1}^{t}\|A^{i-1}\|\,\|B\|\sum_{j=1}^{h}|\sigma_j|^{1/4}\|M_j - M'_j\|\left\|\tilde{W}_{t-i-1:t-i-m}\right\|$$

$$\leq \left(\sum_{i=1}^{t}\kappa^2(1-\gamma)^{i-1}\kappa_B W\sqrt{m}\log^{1/4}\left(\frac{2}{\gamma}\right)\right)\sum_{j=1}^{h}\|M_j - M'_j\|$$

$$\leq W\sqrt{m}\log^{1/4}\left(\frac{2}{\gamma}\right)\left(\frac{\kappa^2\kappa_B}{\gamma}\right)\sum_{j=1}^{h}\|M_j - M'_j\|\,.$$

Using the fact that $\kappa_B\kappa^2/\gamma > 1$, we get a uniform bound:

$$\max\left\{\|\mathbf{x}_t(M_{1:h}) - \mathbf{x}_t(M'_{1:h})\|, \|\mathbf{u}_t(M_{1:h}) - \mathbf{u}_t(M'_{1:h})\|\right\} \leq \frac{\kappa_B\kappa^2 W\sqrt{m}}{\gamma}\log^{1/4}\left(\frac{2}{\gamma}\right)\sum_{j=1}^{h}\|M_j - M'_j\|\,.$$

Using the lipschizness of the cost function from Assumption 3.3, the definition of $\mathcal{K}$, we have

$$
\begin{aligned}
\left|\ell_t(M_{1:h}) - \ell_t(M'_{1:h})\right| &= \left|c_t(\mathbf{x}_t(M_{1:h}), \mathbf{u}_t(M_{1:h})) - c_t(\mathbf{x}_t(M'_{1:h}), \mathbf{u}_t(M'_{1:h}))\right| \\
&\leq \frac{3G\kappa^3 W}{\gamma} \left(\|\mathbf{x}_t(M_{1:h}) - \mathbf{x}_t(M'_{1:h})\| + \|\mathbf{u}_t(M_{1:h}) - \mathbf{u}_t(M'_{1:h})\|\right) \\
&\leq \frac{6G\kappa_B \kappa^5 W^2 \sqrt{m}}{\gamma^2} \log^{1/4}\left(\frac{2}{\gamma}\right) \sum_{j=1}^{h} \|M_j - M'_j\| \, .
\end{aligned}
$$

Finally, we upper bound each $\|M_j - M'_j\|$ by $\|M_{1:h} - M'_{1:h}\|$ to get the result.

$\square$

## B.3 Loss functions with memory

The actual loss $c_t$ at time $t$ is not calculated on $\mathbf{x}_t(M^t_{1:h})$, but rather on the true state $\mathbf{x}_t$, which in turn depends on different parameters $M^i_{1:h}$ for various historical times $i < t$. Nevertheless, $c_t(\mathbf{x}_t, \mathbf{u}_t)$ is well approximated by $\ell_t(M^t_{1:h})$, as stated in Lemma 4.5 and proven next.

*Proof of Lemma 4.5*: By the choice of step size $\eta$, and by the computation of the lipschitz constant of $\ell_t$ w.r.t $M_{1:h}$ in Lemma B.3, we have:

$$
\eta = \frac{2\kappa^3}{L} \sqrt{\frac{2h}{\gamma T}} \, ,
$$

where $L$ is the lipschitz constant of $\ell_t$ w.r.t $M_{1:h}$, computed in Lemma B.3. Thus, for each $j \in [h]$,

$$
\|M^t_j - M^{t-i}_j\| \leq \|M^t_{1:h} - M^{t-i}_{1:h}\| \leq \sum_{s=t-i+1}^{t} \|M^s_{1:h} - M^{s-1}_{1:h}\| \leq i\eta L = 2i\kappa^3 \sqrt{\frac{2h}{\gamma T}} \, .
$$

Observe that $\mathbf{u}_t = \mathbf{u}_t(M^t_{1:h})$. We use the fact proved above to establish that $\mathbf{x}_t$ and $\mathbf{x}_t(M^t_{1:h})$ are close. Observe that $\mathbf{x}_t$ and $\mathbf{x}_t(M^t_{1:h})$ can be written as

$$
\mathbf{x}_t(M^t_{1:h}) = \sum_{i=1}^{t} A^{i-1}\mathbf{w}_{t-i} + \sum_{i=1}^{t} A^{i-1}B \sum_{j=1}^{h} \sigma_j^{1/4} M^t_j \tilde{W}_{t-i-1:t-i-m}\boldsymbol{\phi}_j \, ,
$$

$$
\mathbf{x}_t = \sum_{i=1}^{t} A^{i-1}\mathbf{w}_{t-i} + \sum_{i=1}^{t} A^{i-1}B \sum_{j=1}^{h} \sigma_j^{1/4} M^{t-i}_j \tilde{W}_{t-i-1:t-i-m}\boldsymbol{\phi}_j \, .
$$

Evaluating the difference,

$$
\begin{aligned}
\|\mathbf{x}_t - \mathbf{x}_t(M^t_{1:h})\| &\leq \sum_{i=1}^{t} \|A^{i-1}\|\|B\| \sum_{j=1}^{h} |\sigma_j|^{1/4} \|M^{t-i}_j - M^t_j\| \|\tilde{W}_{t-i-1:t-i-m}\| \\
&\leq 2\kappa^5 \kappa_B W \sqrt{m} \log^{1/4}\left(\frac{2}{\gamma}\right) \sqrt{\frac{2h}{\gamma T}} \sum_{i=1}^{t} i(1-\gamma)^{i-1} \\
&\leq \frac{2\kappa^5 \kappa_B W \sqrt{mh}}{\gamma^{5/2} \sqrt{T}} \log^{1/4}\left(\frac{2}{\gamma}\right) \, .
\end{aligned}
$$

By definition, $\ell_t(M^t_{1:h}) = c_t(\mathbf{x}_t(M^t_{1:h}), \mathbf{u}_t(M^t_{1:h}))$, and by the definition of $\mathcal{K}$ and the projection used in Algorithm 1 we have by Assumption 3.3:

$$
\begin{aligned}
\left|\ell_t(M^t_{1:h}) - c_t(\mathbf{x}_t, \mathbf{u}_t)\right| &= \left|c_t(\mathbf{x}_t(M^t_{1:h}), \mathbf{u}_t(M^t_{1:h})) - c_t(\mathbf{x}_t, \mathbf{u}_t)\right| \\
&\leq \frac{3G\kappa^3 W}{\gamma} \|\mathbf{x}_t(M^t_{1:h}) - \mathbf{x}_t\| \\
&\leq \frac{6G\kappa_B \kappa^8 W^2 \sqrt{mh}}{\gamma^{7/2} \sqrt{T}} \log^{1/4}\left(\frac{2}{\gamma}\right) \, .
\end{aligned}
$$

$\square$

## C  SPECTRAL TAIL BOUNDS

We use the following low-approximate rank property of positive semidefinite Hankel matrices, from Beckermann & Townsend (2016):

**Lemma C.1.** *[Corollary 5.4 in Beckermann & Townsend (2016)] Let $H_n$ be a PSD Hankel matrix of dimension $n$. Then,*

$$\sigma_{j+2k}(H_n) \leq 16 \left[ \exp\left( \frac{\pi^2}{4 \log(8\lfloor n/2 \rfloor/\pi)} \right) \right]^{-2k+2} \sigma_j(H_n) \,.$$

Define the matrix

$$H_m = \int\limits_0^{1-\gamma} \mu_\alpha \mu_\alpha^\top d\alpha \,,$$

where $\mu_\alpha = [1, \alpha, \dots, \alpha^{m-1}] \in \mathbb{R}^m$, and note that $(H_m)_{i,j} = \frac{(1-\gamma)^{i+j-1}}{i+j-1}$. In particular, this is a PSD Hankel matrix of dimension $m$. We prove the following additional properties related to it:

**Lemma C.2.** *Let $\sigma_j$ be the $j^{\text{th}}$ top singular value of $H_m$. Then, for all $T \geq 10$, we have*

$$\sigma_j \leq 156800 \log\left( \frac{2}{\gamma} \right) \cdot \exp\left( -\frac{\pi^2 j}{4 \log T} \right) \leq \frac{1}{2} \log\left( \frac{2}{\gamma} \right) \,.$$

*Proof.* We begin by noting that for any $j$,

$$\sigma_j \leq \text{Tr}(H_m) = \sum_{i=1}^m \frac{(1-\gamma)^{2i-1}}{2i-1} \leq (1-\gamma) \sum_{i=0}^\infty \frac{(1-\gamma)^{2i}}{2i+1} = (1-\gamma)\frac{1}{2} \log\left( \frac{2-\gamma}{\gamma} \right) \leq \frac{1}{2} \log\left( \frac{2}{\gamma} \right) \,.$$

Now, since $T \geq 10$ implies $8\lfloor T/2 \rfloor/\pi > T$, we have by Lemma C.1 that

$$\sigma_{2+2k} \leq \sigma_{1+2k} < 8 \log\left( \frac{2}{\gamma} \right) \cdot \left[ \exp\left( \frac{\pi^2}{2 \log T} \right) \right]^{-k+1}$$

$$< 1120 \log\left( \frac{2}{\gamma} \right) \cdot \exp\left( -\frac{\pi^2 k}{2 \log T} \right) \,.$$

Thus, we have that for all $j$,

$$\sigma_j < 1120 \log\left( \frac{2}{\gamma} \right) \cdot \exp\left( -\frac{\pi^2 (j-2)}{2 \log T} \right) < 156800 \log\left( \frac{2}{\gamma} \right) \cdot \exp\left( -\frac{\pi^2 j}{2 \log T} \right) \,.$$

$\square$

**Lemma C.3.** *For all $T \in \mathbb{N}$ and $0 \leq \alpha \leq 1 - \gamma$, we have:*

1. $\|\mu_\alpha\|^2 \leq 1/\gamma$,

2. $\left| \frac{d}{d\alpha} \|\mu_\alpha\|^2 \right| \leq 2/\gamma^2$.

*Proof.* The first inequality can obtained by evaluating:

$$\|\mu_\alpha\|^2 = \sum_{i=1}^m \alpha^{2i-2} = \frac{1 - \alpha^{2m}}{1 - \alpha^2} \leq \frac{1}{1 - (1-\gamma)^2} \leq \frac{1}{\gamma} \,.$$

To obtain the second inequality, observe that in the summation form, $\left| \frac{d}{d\alpha} \|\mu_\alpha\|^2 \right| = \sum_{i=2}^m (2i - 2)\alpha^{2i-3}$ and hence it monotonically increases with $m$. Thus, if the limit exists for $\left| \frac{d}{d\alpha} \|\mu_\alpha\|^2 \right|$ as

$m \to \infty$ then the limit is the supremum. Evaluating the derivative for the closed form expression of $\|\mu_\alpha\|^2$, and taking the supremum, we get:

$$\left| \frac{d}{d\alpha} \|\mu_\alpha\|^2 \right| \leq \sup_{m \in \mathbb{N}} \left| \frac{(-2m\alpha^{2m-1})(1-\alpha^2) - (1-\alpha^{2m})(-2\alpha)}{(1-\alpha^2)^2} \right|$$

$$= \sup_{m \in \mathbb{N}} \left| \frac{2\alpha - 2m\alpha^{2m-1} + 2m\alpha^{2m+1} - 2\alpha^{2m+1}}{(1-\alpha^2)^2} \right|$$

$$= \lim_{m \to \infty} \left| \frac{2\alpha - 2m\alpha^{2m-1} + 2m\alpha^{2m+1} - 2\alpha^{2m+1}}{(1-\alpha^2)^2} \right|$$

$$= \frac{2\alpha}{(1-\alpha^2)^2} \leq \frac{2(1-\gamma)}{(1-(1-\gamma)^2)^2} \leq \frac{2}{\gamma^2} \, .$$

$\square$

**Lemma C.4.** *Let $\{\phi_j\}$ be the eigenvectors of $H_m$. Then for all $j \in [T]$, $\alpha \in [0, 1-\gamma]$, and $\gamma \leq 2/3$,*

$$|\mu_\alpha^\top \phi_j| \leq \sqrt{\frac{2}{\gamma}} \sigma_j^{1/4} \leq \frac{30}{\sqrt{\gamma}} \log^{1/4}\left(\frac{2}{\gamma}\right) \exp\left(-\frac{\pi^2 j}{16 \log T}\right) \, .$$

*Proof.* Consider the scalar function $g(\alpha) = (\mu_\alpha^\top \phi_j)^2$ over the interval $[0, 1-\gamma]$. First, notice that by definition of $\phi_j$ as the eigenvectors of $H_m$, and $\sigma_j$ as the corresponding eigenvalues, we have

$$\int_0^{1-\gamma} g(\alpha) d\alpha = \int_0^{1-\gamma} \left(\phi_j^\top \mu_\alpha\right)^2 d\alpha = \int_0^{1-\gamma} \phi_j^\top \mu_\alpha \mu_\alpha^\top \phi_j d\alpha = \phi_j^\top Z_h \phi_j = \sigma_j.$$

Since $\|\mu_\alpha\|^2$ is $2/\gamma^2$-lipschitz in the interval $[0, 1-\gamma]$, and $g(\alpha)$ is its projection on $\phi_j$, $g(\alpha)$ is also $2/\gamma^2$-lipschitz. Note that $g(\alpha)$ is also a non-negative function and integrates to $\sigma_j$ over the interval $[0, 1-\gamma]$. Say $R$ is the maximum value achieved by $g(\alpha)$ for all $\alpha \in [0, 1-\gamma]$ then $R \leq \|\mu_\alpha\|^2 \leq 1/\gamma$. Subject to achieveing the maximum at $R$, the non-negative $2/\gamma^2$-lipschitz function over $[0, 1-\gamma]$ with the smallest integral is given by:

$$\Delta(\alpha) = \max\left\{R - \frac{2}{\gamma^2}\alpha, 0\right\} \, ,$$

for which $\int_0^{1-\gamma} \Delta(\alpha) d\alpha = R^2 \gamma^2 / 4$ whenever $\gamma \leq 2/3$. Thus, we get that $R \leq \frac{2}{\gamma}\sqrt{\sigma_j}$ and hence $|\mu_\alpha^\top \phi_j| \leq \sqrt{\frac{2}{\gamma}} \sigma_j^{1/4}$. Using the upper bound on $\sigma_j$ from lemma C.2, we get the result.

$\square$

# D STABILIZED SPECTRAL POLICY

Assumption 3.5 restricts us to competing only against systems for which the zero matrix is $(\kappa, \gamma)$-diagonalizably stable. This implies that there exists a decomposition:

$$A = HLH^{-1},$$

where $\|H\|, \|H^{-1}\| \leq \kappa$, $L$ is diagonal, and $\|L\| \leq 1-\gamma$. However, our proofs only require the bound:

$$\forall \, i \in \mathbb{N}, \qquad \|A^i\| \leq \kappa^2 (1-\gamma)^i,$$

which holds even if $L$ is not diagonal. In fact, it suffices for the zero matrix to be $(\kappa, \gamma)$-strongly stable, as defined in Cohen et al. (2018). For completeness, we recall the definition:

**Definition D.1** (Definition 3.1 in Cohen et al. (2018)). *A linear policy $K$ is $(\kappa, \gamma)$-strongly stable if there exist matrices $L, H$ such that:*

$$A + BK = HLH^{-1},$$

*and the following conditions hold:*

1. *The spectral norm of $L$ is strictly smaller than unity, i.e., $\|L\| \leq 1 - \gamma$.*

2. *The controller and the transformation matrices are bounded, i.e., $\|K\|, \|H\|, \|H^{-1}\| \leq \kappa$.*

However, the assumption of 0 being a $(\kappa, \gamma)$-strongly stable linear controller can be further relaxed by using a precomputed $(\kappa, \gamma)$-strongly stable matrix $K_0$. This can be done using an SDP relaxation as described in Cohen et al. (2018). Given access to a $(\kappa, \gamma)$-strongly stable $K_0$, we learn a *stabilized* spectral policy using online gradient descent defined as follows:

$$\mathbf{u}_t^M := K_0 \mathbf{x}_t^M + \sum_{j=1}^{h} \sigma_j^{1/4} M_j \tilde{W}_{t-1:t-m} \phi_j.$$

Consider playing $\tilde{\mathbf{u}}_t = \mathbf{u}_t + K_0 \mathbf{x}_t$ instead of $\mathbf{u}_t$ at each $t \in [T]$. Observe that the system

$$\mathbf{x}_{t+1} = A\mathbf{x}_t + B\tilde{\mathbf{u}}_t + \mathbf{w}_t, \tag{D.1}$$

when controlled by $\tilde{\mathbf{u}}_t$ behaves the same as the system

$$\mathbf{x}_{t+1} = (A + BK_0)\mathbf{x}_t + B\mathbf{u}_t + \mathbf{w}_t, \tag{D.2}$$

when controlled by $\mathbf{u}_t$. This means that the sequence of states in both the cases is the same. Thus, since the 0 matrix is a $(\kappa, \gamma)-$strongly stable for system (D.2), the regret of our algorithm on system (D.2) is bounded by our result. By the structure of our proofs, for system (D.2), each one of

(i) $\max \left\{ \left\| \mathbf{x}_t^K - \mathbf{x}_t^{K,m} \right\|, \left\| \mathbf{u}_t^K - \mathbf{u}_t^{K,m} \right\| \right\}$,

(ii) $\max \left\{ \left\| \mathbf{x}_t^{K,m} - \mathbf{x}_t^{M^*} \right\|, \left\| \mathbf{u}_t^{K,m} - \mathbf{u}_t^{M^*} \right\| \right\}$,

(iii) $\max \left\{ \| \mathbf{x}_t(M_{1:h}) - \mathbf{x}_t(M'_{1:h}) \|, \| \mathbf{u}_t(M_{1:h}) - \mathbf{u}_t(M'_{1:h}) \| \right\}$,

(iv) $\max \left\{ \| \mathbf{x}_t - \mathbf{x}_t(M_{1:h}^t) \|, \| \mathbf{u}_t - \mathbf{u}_t(M_{1:h}^t) \| \right\}$,

remains bounded. Observe for any state $\mathbf{x}$ and control $\mathbf{u}$, if $\tilde{\mathbf{u}} = \mathbf{u} + K_0 \mathbf{x}$ then:

$$\max \left\{ \|\mathbf{x} - \mathbf{x}'\|, \|\tilde{\mathbf{u}} - \tilde{\mathbf{u}}'\| \right\} \leq \max \left\{ \|\mathbf{x} - \mathbf{x}'\|, \|\mathbf{u} - \mathbf{u}'\| + \kappa \|\mathbf{x} - \mathbf{x}'\| \right\}$$
$$\leq 2\kappa \max \left\{ \|\mathbf{x} - \mathbf{x}'\|, \|\mathbf{u} - \mathbf{u}'\| \right\}.$$

Thus, replacing $\mathbf{u}_t(M)$ with $\tilde{\mathbf{u}}_t(M)$ in (iii) and (iv) yields the same bound with an additional factor of $2\kappa$. Now, observe that $\mathbf{u}^K = K\mathbf{x} = K_0\mathbf{x} + (K - K_0)\mathbf{x} = \tilde{\mathbf{u}}^{(K-K_0)}$. Define

$$\tilde{\mathbf{u}}_t^{K,m} := K_0\mathbf{x}_t + (K - K_0) \sum_{i=1}^{m} (A + BK)^{i-1} \mathbf{w}_{t-i},$$

and choose

$$\tilde{M}_i^* = \sigma_i^{-1/4} (K - K_0) H \left( \sum_{j=1}^{d} \phi_i^\top \boldsymbol{\mu}_{\alpha_j} e_j e_j^\top \right) H^{-1} \qquad \forall\, i \in [h].$$

Now, replacing $\mathbf{u}_t^K$ with $\tilde{\mathbf{u}}_t^{(K-K_0)}$, $\mathbf{u}_t^{K,m}$ with $\tilde{\mathbf{u}}_t^{K,m}$ and $\mathbf{u}_t^{M^*}$ with $\tilde{\mathbf{u}}_t^{\tilde{M}^*}$ in (i) and (ii), we get the same bounds with an additional factor of $2\kappa$. This allows us to conclude that, when competing against the same policy class $\mathcal{S}$, we get an upper bound on the regret with the same order of growth with respect to $T$ and $1/\gamma$.

# E    ADVANTAGE OF SMALLER $\gamma$

Previous works require a stability margin of $\gamma = \Omega(1/polylog(T))$ to ensure an $O(polylog(T))$ running time. In contrast, this work shows that setting $\gamma = \Omega(1/T^k)$ for $k \in (1, 1/12)$ maintains sublinear regret while still guaranteeing an $O(polylog(T))$ running time. In this section, we construct an example demonstrating that choosing $\gamma = 1/T^k$ for $k \in (0, 1/12)$ results in significantly lower aggregate loss compared to $\gamma = 1/polylog(T)$. Consider a noiseless linear dynamical system with parameters $a, b \in \mathbb{R}$, governed by the update equation:

$$x_{t+1} = ax_t + bu_t.$$

The loss function at each time step is defined as:

$$\forall\, t \in [T], \quad c_t(x, u) = \max\{-x, -1\}.$$

Since this is a scalar system, for sufficiently large $\kappa$, the class of $(\kappa, \gamma)$-diagonalizably stable controllers reduces to:

$$S(\gamma) = \{k \in \mathbb{R} \,|\, 0 \le a + bk \le 1 - \gamma\}.$$

If the initial state $x_0 = 1$, then:

$$\min_{k \in S(\gamma)} \sum_{t=1}^{T} c_t(x_t, u_t) = \min_{k \in S(\gamma)} \sum_{t=1}^{T}(-x_t) = -\max_{k \in S(\gamma)} \sum_{t=1}^{T}(a + bk)^{t-1} = -\frac{1 - (1 - \gamma)^T}{\gamma}.$$

Using the fact that $0 \le 1 - \gamma \le e^{-\gamma}$, we can upper and lower bound this expression as:

$$-\frac{1}{\gamma} \le \min_{k \in S(\gamma)} \sum_{t=1}^{T} c_t(x_t, u_t) \le -\frac{1 - e^{-\gamma T}}{\gamma}.$$

Using the lower bound,

$$\min_{k \in S(1/polylog(T))} \sum_{t=1}^{T} c_t(x_t, u_t) \ge -polylog(T),$$

and using the upper bound,

$$\min_{k \in S(1/T^k)} \sum_{t=1}^{T} c_t(x_t, u_t) \le -T^k(1 - e^{-T^{(1-k)}}) \le -T^k/2.$$

Hence, the difference in the minimum costs of the two cases is lower bounded as:

$$\min_{k \in S(1/polylog(T))} \sum_{t=1}^{T} c_t(x_t, u_t) - \min_{k \in S(1/T^k)} \sum_{t=1}^{T} c_t(x_t, u_t) \ge T^k/2 - polylog(T) = \Omega(T^{k/2}).$$

Thus, choosing $\gamma = 1/T^k$ results in a significantly lower cost for the best controller in the policy class $S(\gamma)$ (which we compete against) compared to the case when $\gamma = 1/polylog(T)$. In particular, the improvement is by a polynomial factor in $T$.

## F  USE OF LARGE LANGUAGE MODELS

We used a large language model (ChatGPT) to assist with editing and polishing the writing of this paper. Specifically, the model was used to improve clarity, conciseness, and readability of some sections. All technical content, proofs, algorithms, and experiments were developed entirely by the authors. The model did not contribute to research ideation, discovery, or experimental design.

