# OpenReview forum: "A New Approach to Controlling Linear Dynamical Systems"
_ICLR.cc/2026/Conference — ICLR 2026 Poster_

### Official Review · Reviewer_AjXG · 2025-10-26

**Soundness:** 3
**Presentation:** 2
**Contribution:** 3
**Rating:** 6
**Confidence:** 4

**Summary:**

This paper proposes a new algorithm for online control of linear dynamical systems (LDS) under adversarial disturbances and convex cost functions. The central idea is to use a spectral representation of control policies, transforming the non-convex control problem into a convex online learning problem using Hankel-based spectral filters. This allows efficient computation while preserving strong theoretical guarantees.

**Strengths:**

1. The spectral filters for marginally stable LDS can effectively compress the information of external disturbance. Thus, it allows efficient computation while preserving strong theoretical guarantees.

2. The work sounds solid, although I didn't check the detail proofs.

**Weaknesses:**

1. **The criticism on LQR is not accurate**

For modern industrial applications, the top three control algorithms are likely to be PID, LQR and MPC, despite that those theoretical assumptions are not satisfied. Moreover, the example of drone flying control system is not a good motivate example for the proposed approach. A simple reason is that those scenarios mentioned in this paper are currently handled by the above three algorithms in many commercial products.

Also, the statement about the benefits of marginal stability is misleading: "...yields smoother, more energy-efficient control, useful in settings like robotics, thermal systems, and satellite dynamics". It can lead to small control effort if the open-loop system is unstable or marginally stable. If open-loop is very stable, then one needs extra control effort to reduce the stability margin. Due to its small robustness margin, marginally stability is often not desired in applications related to robotics and satellites. A common setup in those applications is a strong stable tracking controller together with a smooth reference generator. It can yield smooth control input while remaining strong robustness against disturbances.

2. **Some highly related works  are missing**

In Section 1.2, the first two paragraph dates back to the beginning of control theory, which is not necessary. On the other side, it missed some more recent and highly related works called **system level synthesis (SLS)** [1,2] in the control literature. From my understanding, the proposed approach is a special case of SLS, which belongs to a more general framework called **Youla parameterization**. Compared to SLS, the proposed approach has two new features: 1) in SLS setup, the cost function is often known while this work considers the online convex optimization setup; 2) SLS is for general stability while this paper has a tailed algorithm for marginally stable LDS.

3. **Does the proposed approach guarantee stability?**

In this work, it only assumes that the cost function is convex and Lipschitz. Let us consider a linear cost on $x$ and $u$ and there is not constraint $\mathcal{K}$. Then, Algorithm 1 will drive the state to infinity.  Maybe it needs some additional assumptions, e.g., $c(x,u)$ is bounded from below.

4. **Lack of experiments on adversarial cost and disturbance**

This paper claims those benefits in the theory but no experimental illustration is provided. It only investigates simple disturbances including Gaussian,  Rademacher noise and deterministic sinusoidal. And the cost is a fixed quadratic.

[1] Anderson et al. System level synthesis. Annu. Rev. Control, 2019.

[2] Wang et al. A system level approach to controller synthesis. IEEE-TAC, 2019.

**Questions:**

1. Let us consider an extreme case where System (1.1) does not have any external disturbance (i.e. $w_t \equiv 0$). Then, under this setup, Line 7 of Algo. 1 will produce $\tilde{W}=0$. Moreover, whatever $M^t$ is optimized, there is no effective control input, i.e. $u_t=0$. Do those theoretic claims still hold?

2. Follow the previous question. If $w_t$ is small, how will the proposed algorithm behave? Will the loss decreases slower? Or the decay rate does not change?

3. The examples consider simple diagonal $A$ and dense $B$. How about the controllable canonical form where both $A$ and $B$ has certain sparse property? If there is one control input $u\in \mathbb{R}^{1}$ but many states $x\in \mathbb{R}^n$ with large $n$, then the effect on $x(n)$ is delayed by $n$ steps. Will it causing some large oscillations in this loss curve?

---

> ### Author Response · Authors · 2025-11-16
>
> **1. About methods used in practice**
>
> We appreciate the reviewer’s perspective. Our intention was not to comment on the empirical success of LQR, PID, or MPC in practice. Rather, our goal in the introduction was simply to motivate the specific online, adversarial, time-varying problem setting we study. Classical fixed-cost optimal control methods assume a known model and a fixed quadratic objective, while our framework allows arbitrarily varying convex costs and adversarial disturbances. We will revise the wording to make this distinction clearer and to avoid any unintended implication regarding industrial practice.
>
> **2. Marginal Stability**
>
> We thank the reviewer for pointing this out. In our paper, “marginally stable’’ refers only to the closed-loop dynamics of the comparator policy, where the spectral radius of $A + BK$ equals $1 - \gamma$. This parameter appears naturally in the regret bound and is used purely in the theoretical analysis. It is not intended as a recommendation for controller design in engineering applications. We will revise the introduction to clarify that (i) this notion of stability margin pertains solely to the comparator policy in the analysis, and (ii) our discussion concerns the theoretical online-adversarial setting.
>
> **3. Related work**
>
> We thank the reviewer for raising this connection and we will add a short discussion of the SLS literature, including citations to Anderson et al. (2019) and Wang et al. (2019). At a high level, SLS and Youla provide parameterizations for stabilizing controllers in offline fixed-cost optimal control, whereas our work focuses on online convex costs, adversarial disturbances, and regret guarantees relative to a stable comparator. The spectral filters we use are universal basis functions and are constructed independently of $(A,B)$. We will clarify these distinctions in the revision.
>
> **4. Linear costs**
>
> We appreciate the opportunity to clarify this point. In our framework, the control input $u_t$ is always chosen from a fixed compact convex set $\mathcal K$ (Algorithm 1, line 8; Definition 4.2). Thus, the unconstrained scenario described in the comment does not arise.
> In addition, our analysis is comparator-based: if the comparator policy $K$ induces $(\kappa,\gamma)$-stable closed-loop dynamics, then Algorithm 1 achieves regret $O(\gamma^{-4}\sqrt{T})$ relative to this stable reference. As a result, the algorithm cannot diverge while maintaining sublinear regret, because divergence would incur unbounded cost relative to the bounded cost of the comparator. We will clarify this in the revision.
>
> **5. Experiments on adversarial cost and disturbance.**
>
> Thank you for this comment. In adversarial online control, fully adaptive adversaries can force arbitrary trajectories, so worst-case behavior is typically handled analytically rather than through visualization. This is standard in both the online control and adversarial OCO literature, where experiments focus on representative stochastic or structured disturbances (Gaussian, Rademacher, sinusoidal, etc.). Our experiments follow this convention. We will clarify this context in the revision.
>
> **Responses to the Reviewer’s Questions**
>
> *Q1. “If $w_t = 0$, the controller outputs zero. Do the claims still hold?”*
>
> Yes. With $x_0 = 0$ (absorbed into $w_0$), both the comparator and Algorithm 1 apply zero control and remain at the zero state when $w_t = 0$. The comparator’s cumulative loss is $\sum_t c_t(0,0)$, and the algorithm incurs the same loss. Thus, the regret guarantee holds immediately. The disturbance-free case is a straightforward special instance covered by the theory.
>
> *Q2. “If $w_t$ is small, what happens to the loss decay?”*
>
> Assumption 3.2 ensures $|w_t| \le W$. The constants in Theorem 2.1 depend on $W$ only through terms proportional to $W^2$. Thus, smaller disturbances lead to proportionally smaller cumulative loss, while the regret rate $O(\gamma^{-4}\sqrt{T})$ remains unchanged. We will add a brief clarification of this monotonic dependence.
>
> *Q3. “What about canonical forms with delayed controllability?”*
>
> We appreciate the question. Different coordinate representations can make the effect of the control input appear delayed on certain coordinates, but this does not change the system’s controllability or any aspect of our guarantees. Our analysis already covers all controllable linear systems, including underactuated settings such as $u \in \mathbb R^1$ and $x \in \mathbb R^n$, which are also included in our experiments. If the reviewer had a specific construction in mind, we would be happy to clarify further.

---

> > ### Comment · Reviewer_AjXG · 2025-11-27
> >
> > Thank the authors for detail response. Most of my questions have been addressed. Just a few minor follow up comments.
> >
> > - Point 1
> >
> > First, classic optimal control methods require a known model. I guess the proposed method also need a known model; otherwise, it is impossible to compute $w_t$. Second, conventional control methods (e.g. MPC) can handle arbitrary cost function (convex or non-convex). Here the key difference is that classic control assumes the cost function is available for prediction while in the online setup, the cost function is only revealed at each step.
> >
> > - Point 5
> >
> > I understand that the most important part is the theoretical result. But it will be more convincing to show some adversarial attack results rather than some fixed disturbance patterns. It is not that hard. Just treat your algorithm as a differentiable simulator and apply PGD attack to learn $w_t$ online. Hope that I can see some empirical results in your future work, not necessary in the revise version.
> >
> > - Q1
> >
> > The reason I ask this question is that Algo.1 does not mention the initialization of $w_{t-m},\ldots, w_{t-1}$ as well as $x_t$ when $t=0$.

---

> > > ### Author Response · Authors · 2025-11-28
> > >
> > > Thank you for the follow up comments, we hope the following explanations will clarify those points:
> > >
> > >
> > > 1. First of all regarding known model: yes, as we explicitly state in the first page of the paper (last sentence) we know $A,B$ and thus as the reviewer mentioned they appear in our Algorithm 1. As we responded to other reviewer’s concerns here, it is possible to first estimate $A,B$ as in Chen & Hazan (2021) and then use our approach for the separate control phase. This doesn’t affect our results and we will make the connection more clear in the revised version.
> > >
> > > We thank the reviewer for their comment about non quadratic cost functions. Indeed, the novelty of our work is not only in extending to more general cost functions, but rather to general online cost functions - the work of [1] already considered online quadratic, and as the reviewer mentioned other classical methods considered static convex. We will make sure to add those to the related work as well in the revised version.
> > >
> > > We agree that numerical MPC methods can heuristically handle non-convex cost functions using local optimization techniques. However, we emphasize that non-convex optimization is generally NP-hard. Without a guarantee of finding the global minimum, it is impossible to establish standard regret bounds, which require comparing the algorithm's performance against a globally optimal comparator.
> > >
> > > While extensions to local regret or oracle-based non-convex settings exist in the online learning literature, our work focuses on the convex relaxation of the control problem. We highlight that even with simple convex cost functions (e.g., quadratic), the optimization landscape with respect to the linear controller is non-convex. By utilizing improper learning and our novel convex relaxation, we convert this into a tractable problem. Thus, we target the most general setting where rigorous global regret guarantees are computationally feasible.
> > >
> > > 2. This is an interesting experimental idea which is fully covered by our theoretical guarantees. However, the motivation to study online learning is not necessarily to achieve adversarial training, but rather to have a guarantee for any possible environment. As the reviewer suggested, we leave such adversarial attack experiments for future work. Our experiments already cover both stochastic and deterministic disturbances (Gaussian, Rademacher, sinusoidal), and indeed, the focus of this work is a new approach with a provable guarantee using novel techniques.
> > >
> > > 3. In section 3 (prelimenaries) we explicitly write $w_t=0$ for any $t<0$ and that $x_t=0$ WLOG as the disturbances are bounded. We will consider adding this as input to algorithm 1 as well.
> > >
> > >
> > > References
> > >
> > > [1] Xinyi Chen and Elad Hazan (2021), “Black-Box Control for Linear Dynamical Systems”
> > >
> > > [2] Alon Cohen, Avinatan Hassidim, Tomer Koren, Nevena Lazic, Yishay Mansour, and Kunal Talwar. Online linear quadratic control, 2018.

---

### Official Review · Reviewer_tXpd · 2025-10-28

**Soundness:** 3
**Presentation:** 3
**Contribution:** 2
**Rating:** 6
**Confidence:** 3

**Summary:**

The paper proposes a new \emph{spectral convex relaxation} for online (adversarial) control of linear dynamical systems.
It replaces explicit system-dependent features with a fixed, universal set of \emph{Hankel eigenvector filters},
thereby transforming control into an online convex optimization problem.
The resulting algorithm achieves a regret bound of $\widetilde{\mathcal{O}}(\gamma^{-4}\sqrt{T})$,
where $\gamma$ denotes the \emph{stability margin} of the comparator policies (i.e., the spectral radius bound of the closed-loop matrix).
This dependency in $\gamma$ is better compared to existing approaches.
The method runs in polylogarithmic time per step and is supported by a detailed approximation analysis
and experiments demonstrating a competitive runtime efficiency compared to existing baselines.

**Strengths:**

The paper is very well written and the overall presentation is clear and easy to follow.
The structure and flow of ideas make it straightforward to understand the motivation,
technical setup, and implications of the proposed approach.
Both the algorithmic framework and the main theoretical result Theorem 2.1) are presented in a transparent way,
with all relevant quantities and assumptions clearly defined.
In particular, the paper explicitly spells out the concrete hyperparameter choices
$m$, $h$, and $\eta$ that lead to the stated regret bound,
which greatly improves the reproducibility and interpretability of the theoretical results.
A further strength is the improved dependence on the stability margin $\gamma$ in the regret bound,
which represents a significant advance over previous approaches whose computational or regret guarantees scale more unfavorably with $\gamma$.
Finally, the discussion of related work is comprehensive and well organized,
situating the contribution within the broader literature on online control and learning-based methods.

**Weaknesses:**

**Weaknesses and Suggestions.**

 1) *Reliance on known $(A,B)$ is unrealistic in many applications.*
  The entire pipeline assumes exact knowledge of the system matrices to reconstruct disturbances and build spectral features.
  This limits applicability to model-mismatch or unknown-dynamics settings.
  Suggestion: Add a discussion (or short appendix) on robustness to misspecified $(A,B)$ and outline a data-driven variant where
  $(\widehat A,\widehat B)$ are estimated online (or pre-estimated) with explicit stability conditions under identification error,
  e.g., bounds of the form $\|A-\widehat A\|,\|B-\widehat B\|\le\varepsilon$ and their impact on regret constants.

  2) *Placement of Theorem 2.1 (many forward references).*
  As written, Theorem 2.1 appears before the technical setup of Section 3, causing heavy forward referencing and making the statement hard to parse on first reading.
  Suggestion: Move Theorem 2.1 to immediately \emph{after} Section 3, once the comparator class, spectral filters, and surrogate loss are fully defined.
  This will make the result self-contained at its first occurrence.

  3) *Reorder Section 3 before the main results.*
  Closely related to the previous point: placing the entire Section~3 (setup, assumptions, and construction of the spectral controller)
  before the main theorem would improve readability and reduce back-and-forth navigation.
  Suggestion: New order: (i) problem setup and assumptions, (ii) spectral feature construction \& surrogate loss (current Section 3),
  (iii) then algorithm and main regret theorem.

  4) *No empirical verification of the $\sqrt{T}$ regret scaling in Theorem 2.1.*
  The current experiments do not test the $\sqrt{T}$ dependence, so they cannot confirm the rate empirically.
  Suggestion (add a minimal experiment):

   4a) Fix a stable LTI instance and cost sequence within the paper's assumptions (convex Lipschitz costs), and fix a stability margin $\gamma$ (e.g., by choosing a comparator with spectral radius $1-\gamma$).

4b) Run the proposed method for horizons $T\in\{2^{10},2^{11},\dots,2^{18}\}$, keeping all other hyperparameters at the prescribed values in Theorem 2.1 (the stated $m,h,\eta$ schedule).

4c) For each $T$, compute regret with respect to the comparator class used in the theory (or as close as feasible), average over multiple seeds, and report the mean $\pm$ std.

4d) Plot $\log(\text{regret})$ vs.\ $\log T$ and report the fitted slope (expectation: slope $\approx\frac{1}{2}$ within confidence bands).
    \item (Optional) Repeat for a few $\gamma$ values to visualize the $\gamma$-dependence of constants in the regret (even if asymptotically the slope stays $\approx\frac{1}{2}$).

**Questions:**

1) What can be said when $A,B$ are unknown (cf. weakness section above)?
    Do the results extend to estimated $(\widehat A,\widehat B)$, and how do identification errors
    $\|A-\widehat A\|$, $\|B-\widehat B\|$ affect the feature construction, stability, and regret guarantees?

2) When referring to a ``convex cost,'' do you mean convex jointly in $(x,u)$ for each $t$?
    Please clarify the precise convexity and Lipschitz assumptions, including in which variables and norms they hold.

3) In the first displayed equation of Section 1.1, the policy class in the regret definition is missing.
    Please specify explicitly over which set of policies or feedback gains the infimum is taken.

4) In Algorithm 1, the loss function $\ell_t$ is referenced but not defined at that point.
    It only appears later in Equation (4.1), which interrupts the reading flow.
    Please either define $\ell_t$ where Algorithm~1 appears or add a forward reference in the algorithm caption.

5) Definition 3.1: can the stated condition occur at any time $t$?
    I assume yes, please make the time quantification explicit.

6) Definition 3.4: please be precise that $K$ denotes a feedback gain that induces a linear policy;
    $K$ itself is not a policy. Consider rephrasing accordingly to avoid ambiguity.

---

> ### Author Response · Authors · 2025-11-15
>
> We thank the reviewer for the positive assessment of the paper’s clarity, theoretical soundness, and improved dependence on the stability margin. We appreciate the detailed stylistic suggestions and address each point below.
>
> **1. Known System Matrices**
> We agree that the theoretical analysis assumes known system matrices, consistent with the standard formulation in nonstochastic control. However, the approach naturally extends to the unknown-dynamics or mildly misspecified setting. One may first estimate $(A,B)$ using standard system-identification methods and then apply the proposed spectral-filter controller to the estimated model, as in Chen and Hazan (2021, "Black-Box Control for Linear Dynamical Systems"). Our contribution strictly improves the control phase of such black-box pipelines, yielding sharper dependence on the stability margin and exponentially faster runtime, while the estimation phase and its robustness guarantees remain unchanged. Consequently, identification errors (e.g., bounds of the form $|A - A^\star| \le \epsilon_A$ and $|B - B^\star| \le \epsilon_B$) follow the same guarantees as in prior work. We will add a brief discussion of this robustness and the connection to data-driven extensions in the revision.
>
> **2. Paper Organization (Placement of Theorem 2.1)**
> We appreciate the reviewer’s stylistic suggestion. Presenting the main theorem early is a common convention in theoretical work, as it provides readers with a clear statement of the goal and the structure of the result before entering the technical development. This helps orient the reader and allows subsequent sections to be read with the main result in mind. For this reason, we prefer to retain the current organization.
>
> **3. Empirical Regret Scaling**
> We thank the reviewer for the suggestion. Empirically evaluating regret in online control is challenging because computing the optimal policy is NP-hard outside of very restricted settings. As a result, prior work in nonstochastic control evaluates loss trajectories rather than regret, which is the protocol we follow in our experiments. These evaluations already illustrate performance trends as a function of both the horizon and the stability margin (Section 5.4). A full regret-scaling study across multiple horizons would require substantial additional experimentation, and the ICLR reviewing guidelines encourage keeping rebuttal-phase experiments limited in scope. We therefore focus on the existing loss-trajectory evaluations in Section 5.4 and will make these trends clearer in the revision.
>
> **4. Technical Clarifications**
> We thank the reviewer for highlighting these clarifications. Each cost function $c_t(x,u)$ is convex and Lipschitz in both arguments with respect to the Euclidean norm, and we will restate this explicitly. In Section 1.1, we will make the comparator class precise by directly referencing Definition 3.4. In Algorithm 1, we will add a forward reference to Eq. (4.1), where $\ell_t$ is formally defined. In Definition 3.1, the time index simply denotes the update step, and the phrase “can be steered to any target state from any initial state” means that there exists some finite sequence of inputs achieving this transfer; there is no requirement on the sequence length. We will consider rephrasing the definition to make this fully explicit. Finally, in Definition 3.4, we will clarify that $K$ denotes a linear feedback gain inducing the policy $u_t = K x_t$.

---

> > ### Comment · Reviewer_tXpd · 2025-11-26
> > **Response to rebuttal**
> >
> > I thank the authors for clarifying my questions.

---

### Official Review · Reviewer_Enzf · 2025-10-29

**Soundness:** 3
**Presentation:** 2
**Contribution:** 2
**Rating:** 6
**Confidence:** 2

**Summary:**

This paper proposes an online control algorithm for linear stochastic systems with adversarial disturbance. The proposed algorithm approximates a policy using spectral features derived from Hankel matrix eigenvectors, allowing efficient updates with regret comparable to existing approaches while significantly improving runtime. The author demonstrates that the proposed algorithm can match or surpass existing baselines empirically while using far fewer parameters.

**Strengths:**

- The use of spectral filtering in online control appears to be a novel idea that could have a broader impact on the control domain.
- The theoretical analysis is thorough and clearly demonstrates improvement over prior approaches.

**Weaknesses:**

- The empirical evaluation remains relatively narrow in scope. Including results on higher-dimensional systems would strengthen the paper.
- While the main advantage of the proposed algorithm lies in its runtime efficiency, this aspect is not evaluated in the experiments.

**Questions:**

- Could the authors provide more intuition behind the motivation for using spectral filters? What advantages do spectral features offer compared to other feature representations in online control?
- Could the authors comment on the actual runtime improvement over baseline methods in the experiments?

---

> ### Author Response · Authors · 2025-11-15
>
> We thank the reviewer for the positive assessment. Below we address the remaining points.
>
> **1. Scope of Empirical Evaluation**
> We appreciate the suggestion to study higher-dimensional systems. Our current paper already includes such an analysis in Section 5.2 ("Ablating State Dimensionality"), where we vary the state dimension $d \in {2,5,10,20}$ and report normalized loss in Fig. 6. The control input dimension is fixed at $n = 1$, which makes the problem increasingly challenging as $d$ grows — substantially more difficult than the case $d \approx n$. The goal of this setup is precisely to demonstrate that OSC remains effective even when the state dimension is much higher than the controller dimension. We will consider extending the experiments to larger values of $d$.
>
> **2. Runtime Efficiency**
> We agree that explicit runtime benchmarks would be valuable. The core efficiency insight, however, already appears in Section 5.3 ("Ablating Number of Parameters in OSC") and Fig. 4, which show that OSC matches GPC performance while using an order of magnitude fewer parameters (e.g., $h = 20$ vs.\ $m = 50$). Since both algorithms scale linearly in their number of parameters, this reduction directly translates into proportional savings in runtime and memory.
>
> We did not include wall-clock comparisons because no public implementation of the fast online convolution method of Agarwal et al. (2023) is available, and in the absence of such an implementation a direct timing comparison would not be meaningful. Since this is a theoretical paper, we chose to provide preliminary experiments that focus on parameter efficiency, which already reflects the core efficiency advantage of OSC in a fair and implementation-independent manner.
>
> We will make this clarification explicit in the experiments section and emphasize that Corollary 2.2 already formalizes the theoretical runtime improvement.
>
> **3. Intuition for Spectral Features**
> We thank the reviewer for the question. As discussed in Section 2, the top Hankel eigenvectors form an orthogonal basis that captures the dominant impulse-response modes induced by the optimal linear policy, effectively compressing them on a logarithmic scale. Projecting disturbances onto these spectral filters provides a compact feature representation that closely approximates the behavior of the optimal linear controller, thereby enabling improper learning of a convex policy with far fewer parameters. We will make this intuition more explicit in the revision.

---

### Official Review · Reviewer_71FK · 2025-11-01

**Soundness:** 3
**Presentation:** 3
**Contribution:** 2
**Rating:** 6
**Confidence:** 4

**Summary:**

This paper considers the problem of controlling a known linear dynamical system under possibly adversarial disturbances and time-varying costs. The authors propose an approach motivated by "spectral filters", wherein the eigenvalues/eigenvectors of a particular Hankel matrix determined by a user-set stability margin (over)estimate are used to parameterize a control law that maps past disturbances to controls. Compared to prior work in "non-stochastic control", by using this particular parameterization, the proposed method attains regret bounds that improve the dependence on the stability margin, as well as improve the run-time dependence on the stability margin from polynomial to polylogarithmic.

**Strengths:**

Overall, this paper is well-written and the message is rather clear. The proposed method is a novel adaptation of prior literature in both non-stochastic control and spectral filtering that seems to improve existing regret and runtime guarantees for this problem under relatively relaxed theoretical conditions. The approximation techniques involving the Hankel matrix are interesting, and as far as I know rather novel in the space of (linear) control. Thus, I think this paper contains worthy and interesting content for the niche of learning on dynamical systems researchers.

**Weaknesses:**

Some immediate points that deserve some discussion are as follows:

1. The results seem to rely on the dynamics matrices $A,B$ being known. How well do the proposed results extend to the unknown dynamics case?

2. The proposed method seems rather tied to linear dynamics parameterization. How does one naturally apply this method to non-linear dynamics/policy parameterization? On that note, the discussion of applying GPC and OSC to nonlinear systems (with potentially nonlinear parameterization) in experiment Section 5.1 is rather terse, and it is not immediately clear how the proposed methods are being applied---the appendix doesn't seem to have further information on this front.

3. The numerical experiments lack comparisons to even simple baselines. For example, considering some of the experiments concern control with Gaussian or zero-mean stochastic disturbances, it would be instructive to see the loss dynamics of, e.g. online LQR. Furthermore, in the known-linear-dynamics (potentially quadratic-loss case), there is quite a lot of literature on robust control, see e.g. classical mixed H2/H$\infty$ control, competitive control [1], and adversarially robust synthesis [2]. I think it is both worth including some discussion about robust control perspectives, and a couple more simple baselines beyond disturbance-action filter approaches to get a better sense of the method's relative performance.

4. As a minor comment, it should be noted the margins of the submission seem to have been different than the template.

[1] Goel and Hassibi, "Competitive Control"

[2] Lee et al. "Performance-Robustness Tradeoffs in Adversarially Robust Linear-Quadratic Control"

**Questions:**

Please see under Weaknesses.

---

> ### Author Response · Authors · 2025-11-15
>
> We thank the reviewer for the thoughtful and constructive feedback. We address each point below.
>
> **1. Unknown System Matrices**
>
> In the unknown-dynamics setting, one could first perform system identification and then apply our spectral-filter control method. This is fully compatible with existing black-box pipelines, e.g., Chen & Hazan (2021), “Black-Box Control for Linear Dynamical Systems”, where a system estimation phase is followed by a non-stochastic control phase. Our contribution strictly improves the control phase, while leaving the estimation phase unchanged. Hence the approach extends naturally to the unknown-dynamics case without modification, and we will make this point explicit in the revision.
>
> **2. Nonlinear Dynamics and Policy Parameterization**
>
> We thank the reviewer for raising this point. Our paper already includes an explicit nonlinear extension and corresponding experiments (§5.1). Specifically, we evaluate both linear and nonlinear systems (LDS ReLU), and compare linear policies to two-layer ReLU networks trained on the same spectral features. The nonlinear setting therefore uses a nonlinear policy parameterization operating on filtered inputs — demonstrating that our spectral representation is not limited to linear control laws. As shown in Fig. 5, these spectral features continue to provide benefits when coupled with expressive nonlinear models, confirming their compatibility with richer architectures (e.g., MLPs, CNNs, Transformers). We will make this clearer in the revision by explicitly noting that the LDS ReLU experiments correspond to nonlinear dynamics and nonlinear policy parameterization.
>
> Formally, the nonlinear variant simply replaces the linear projection of filtered disturbances with a neural mapping. That is, instead of computing the linear combination
> $u_t = \sum_{i=1}^h \sigma_i^{1/4} M_i \tilde W_{t-1:t-m} \phi_i,$
> we feed the same filtered features
> $z_t = [,\sigma_i^{1/4} \tilde W_{t-1:t-m} \phi_i,]_{i=1}^h$
> into a neural network, yielding a nonlinear controller
> $u_t = f\theta(z_t).$
> The input dimension is significantly smaller for the spectral model, while achieving the same performance.
>
> Moreover, $w_t$ can be computed online in the same way as in the linear system case, using a noiseless simulator that predicts the next state given the current state and control input. Let sim denote such a simulator, so that it returns the predicted next state in the absence of noise. Then
> $w_t = x_{t+1} - \text{sim}(x_t, u_t).$
>
> **3. Experimental Baselines and Robust-Control Discussion**
>
> We appreciate the reviewer’s suggestions and will consider including the following in the revision: (i) Online LQR and $H_\infty$ robust control (DARE-based) as additional baselines, and (ii) a discussion connecting our regret formulation to competitive control (Goel & Hassibi 2022) and adversarially robust LQ synthesis (Lee et al. 2023).
> We also note that our algorithm applies to arbitrarily changing convex cost functions, while both Online LQR and $H_\infty$ robust control assume fixed quadratic costs and therefore cannot be used in the more general cost settings considered in this paper.
>
> **4. Minor Formatting**
>
> We thank the reviewer for pointing out the possible inconsistency and will make sure our margin is consistent in the final revision.

---

### Meta-Review · Area_Chair_G4pA · 2026-01-05

**Summary:**

This paper introduces spectral filtering to improve the computational efficiency of online nonstochastic control.

**Reviewer Concerns:**

Concerns were mainly around presentation (introducing theorem before assumptions), requirements of known dynamics, poor attribution of related work, and lack of comparison to simple experimental baselines. The authors promised to right 3/4 of these, instead stating that some of their presentation choices were a matter of presference.

**Reviewer Scores:**

I believe that one or two of the reviewers would increase their score, but most felt only slightly positive.

---

### Decision · Program_Chairs · 2026-01-26

Accept (Poster)